



**1  Examination of aerosol impacts on convective clouds and precipitation in two**
**2  metropolitan areas in East Asia; how varying depths of convective clouds between**
**3  the areas diversify those aerosol effects?**

Seoung Soo Lee[1,2], Jinho Choi[3], Goun Kim[4], Kyung-Ja Ha[2,5,6], Kyong-Hwan Seo[3], Junshik
Um[3], Youtong Zheng[7]
[1]Earth System Science Interdisciplinary Center, University of Maryland, Maryland
[2]Research Center for Climate Sciences, Pusan National University, Busan, Republic of
Korea
[3]Department of Atmospheric Sciences, Division of Earth Environmental System, Pusan
National University, Busan, Republic of Korea
[4]Marine Disaster Research Center, Korea Institute of Ocean Science and Technology,
Pusan, Republic of Korea
[5]Center for Climate Physics, Institute for Basic Science, Busan, Republic of Korea
[6]BK21 School of Earth and Environmental Systems, Pusan National University, Busan,
Republic of Korea
[7]The Program in Atmospheric and Oceanic Sciences, Princeton University,
and National Oceanic and Atmospheric Administration/Geophysical Fluid Dynamics
Laboratory, Princeton, New Jersey, USA
Corresponding author: Seoung Soo Lee
Office: (303) 497-6615
Fax: (303) 497-5318
E-mail: cumulss@gmail.com, slee1247@umd.edu



## Abstract

This study examines the role played by aerosols in the development of clouds and precipitation in two metropolitan areas in East Asia that has experienced substantial increases in aerosol concentrations over the last decades. These two areas are the Seoul and Beijing areas and the examination has been done by performing simulations using a cloud-system resolving model (CSRM). Aerosols are advected from the continent to the Seoul area and this increases aerosol concentrations in the Seoul area. These increased aerosol concentrations induce the enhancement of condensation that in turn induces the enhancement of deposition and precipitation amount in a system of less deep convective clouds as compared to those in the Beijing area. In a system of deeper clouds in the Beijing area, increasing aerosol concentrations also enhance condensation but reduce deposition. This leads to aerosol-induced negligible changes in precipitation amount. Also, in the system, there is a competition for convective energy among clouds with different condensation and updrafts. This competition results in different responses to increasing aerosol concentrations among different types of precipitation, which are light, medium and heavy precipitation in the Beijing area. In both of the areas, aerosol-induced changes in freezing play a negligible role in aerosol-precipitation interactions as compared to the role played by aerosol-induced changes in condensation and deposition.





## 1. Introduction

With increasing aerosol loading or concentrations, cloud-particle sizes and autoconversion,
which represent cloud microphysical properties, can be changed. Autoconversion is a
process where cloud-liquid particles or droplets grow to form raindrops via collision and
collection among droplets. In general, with increasing particle sizes, the efficiency of
collision and collection among droplets increases. Increasing aerosol loading is known to
make the particle size smaller and thus make the efficiency of collision, collection and
autoconversion lower. This leads to less cloud liquid which can grow to be raindrops and
there is more cloud liquid present in the air to be evaporated or frozen. Studies have shown
that increases in cloud-liquid mass due to increasing aerosol loading can enhance the
freezing of cloud liquid and parcel buoyancy, which lead to the invigoration of convection
(Rosenfeld et al., 2008; Fan et al., 2009). Via the invigoration of convection, precipitation
can be enhanced. The dependence of aerosol-induced invigoration of convection and
precipitation enhancement on aerosol-induced increases in condensational heating in the
warm sector of a cloud system has been shown (e.g., van den Heever et al., 2006; Fan et
al., 2009; Lee et al., 2018). Increasing cloud-liquid mass induces increasing evaporation,
which intensifies gust fronts. This in turn strengthens convective clouds and increases the
amount of precipitation (Khain et al., 2005; Tao et al., 2007; Storer et al., 2010; Tao et al.,
2012; Lee et al., 2017; Lee et al., 2018). It is notable that aerosol-induced precipitation
enhancement is strongly sensitive to cloud types that can be defined by cloud
characteristics such as cloud depth (e.g., Tao et al., 2007; Lee et al., 2008; Fan et al., 2009).
Since East Asia was industrialized, there have been substantial increases in aerosol
concentrations over the last decades in East Asia (e.g., Lee et al., 2013; Lu et al., 2011; Oh
et al., 2015; Dong et al., 2019). These increases are far greater than those in other regions
such as North America and Europe (e.g., Lu et al., 2011; Dong et al., 2019). While those
increasing aerosols affect clouds, precipitation and hydrologic circulations in the
continental East Asia, the increase in the advected aerosols from the continent to the
Korean Peninsula affect clouds, precipitation and hydrologic circulations in the Korean
Peninsula (Kar et al., 2009). This study aims to examine effects of the increasing aerosols
and their advection on clouds and precipitation in East Asia. As a first step to this


examination, this study focuses on two metropolitan areas in East Asia which are the
Beijing and Seoul areas. The population of each of the Beijing and Seoul areas is ~ 20
millions. Associated with this, these areas have lots of aerosol sources (e.g., traffic) and
have made a substantial contribution to the increases in aerosol concentrations in East Asia.
Hence, we believe that these two cities can represent overall situations related to increasing
aerosol concentrations in East Asia.
As mentioned above, aerosol-cloud interactions (and their impacts on precipitation) are
strongly dependent on cloud types and thus to gain a more general understanding of those
interactions, we select cases from the Beijing and Seoul areas with different cloud types.
A selected case from the Beijing area involves deep convective clouds that reach the
tropopause, while a selected case from the Seoul area involves comparatively shallow (or
less deep) convective clouds. Via comparisons between these two cases, we aim to identify
mechanisms that control varying aerosol-cloud interactions with cloud types.
To examine impacts of aerosols on clouds and precipitation in the cases, numerical
simulations are performed, as a way of fulfilling above-described aim. These simulations
use a cloud-system resolving model (CSRM) that has reasonably high resolutions to
resolve cloud-scale processes that are related to cloud microphysics and dynamics. Hence,
these simulations are able to find process-level mechanisms in association with cloud-scale
processes.

**2. Case description**

In the Seoul area, South Korea, there is an observed mesoscale convective system (MCS)
for a period from 03:00 LST (local solar time) to 18:00 LST December 24th 2017. During
this period, there is a recorded moderate amount of precipitation and its maximum
precipitation rate reaches ~ 13 mm hr$^{-1}$. At 21:00 LST December 23rd 2017, synoptic-scale
features develop in favor of the formation and development of the selected MCS and
associated moderate rainfall. A low-pressure trough was over northeast China and the
Yellow Sea (Figure 1a). Along the flank of the low-pressure system, there was the
southwesterly low-level jets to transport warm and moist air. This warm and moist air is
originated from the Yellow Sea and transported to the Korean Peninsula (Figure 1a). The



southwesterly low-level jet plays an important role in the formation and development of
rainfall events in the Korean Peninsula by fetching warm and moist air (Hwang and Lee
1993; Lee et al. 1998; Seo et al. 2013; Oh et al. 2018).

There was another observed MCS case in the Beijing area, China for a period from

14:00 LST on July 27th to 00:00 LST July 28th 2015. There is a substantial recorded amount
of precipitation for this period and its maximum precipitation rate reaches ~ 45 mm hr$^{-1}$.
At 09:00 LST July 27th 2015, synoptic-scale features develop in favor of the formation and
development of the selected MCS. The southerly low-level jet forms and develops heavy
rainfall events in the Beijing area by transporting warm and moist air to the area (Figure
1b).

**3.   CSRM and simulations**


**3.1 CSRM**


The Advanced Research Weather Research and Forecasting (ARW) model (version 3.3.1)
is used as a CSRM. The ARW model is a compressible model with a nonhydrostatic status.
A 5th-order monotonic advection scheme is used to advect microphysical variables (Wang
et al., 2009). The Rapid Radiation Transfer Model (RRTMG; Mlawer et al., 1997; Fouquart
and Bonnel, 1980) is adopted to parameterize shortwave and longwave radiation in
simulations. A microphysics scheme that is used in this study calculates the effective sizes
of hydrometeors that are fed into the RRTMG, and the RRTMG simulates how these
effective sizes affect radiation.

The CSRM adopts a bin scheme as a way of parameterizing microphysics. The

Hebrew University Cloud Model (HUCM) detailed in Khain et al. (2011) is the bin scheme.
A set of kinetic equations is solved by the bin scheme to represent size distribution
functions for each class of hydrometeors and aerosols acting as cloud condensation nuclei
(CCN). The hydrometeor classes are water drops, ice crystals (plate, columnar and branch
types), snow aggregates, graupel and hail. There are 33 bins for each size distribution in a
way that the mass of a particle $m_j$ in the j bin is to be $m_j = 2m_{j-1}$.



**3.2 Control runs**


For a three-dimensional CSRM simulation of the observed case of convective clouds in the
Seoul (Beijing) area, i.e., the control-s (control-b) run, a domain just over the Seoul
(Beijing) area, which is shown in Figure 2a (2b), is used. This domain adopts a 300-m
resolution. The control-s run is for a period from 03:00 to18:00 LST December $24^{th}$ 2017,
while the control-b run is for a period from 14:00 LST on July $27^{th}$ to 00:00 LST July $28^{th}$
2015. The length of the domain is 170 (140) km in the east-west (north-south) direction for
the control-s run, and 280 (240) km for the control-b run. There are 100 vertical layers and
these layers employ a sigma coordinate that follows the terrain. The top pressure of the
model is 50 hPa for both of the control-s and control-b runs.  On average, the vertical
resolution is ~200 m.

Reanalysis data, which represent the synoptic-scale features, provide initial and

boundary conditions of variables such as wind, potential temperature, and specific
humidity for the simulations. The Met Office Unified Model (Brown et al., 2012) produces
these data every 6 hours with a 0.11° × 0.11° resolution. The simulations adopt an open
lateral boundary condition. The Noah land surface model (LSM; Chen and Dudhia, 2001)
calculates surface heat fluxes.

The current version of the ARW model is not able to consider the spatiotemporal

variation of aerosol properties. In order to take into account the spatiotemporal variation of
aerosol properties, which is typical in metropolitan areas, such as composition and number
concentration, an aerosol preprocessor, which is able to consider the variability of aerosol
properties, is developed and used in the simulations. This aerosol preprocessor interpolates
or extrapolates background aerosol properties in observation data such as aerosol mass
(e.g., $PM_{2.5}$ and $PM_{10}$) into grid points and time steps in the model. PM stands for
particulate matter. The mass of aerosols with diameter smaller than 2.5 (10.0) μm per unit
volume of the air is $PM_{2.5}$ ($PM_{10}$).

$PM_{2.5}$ or $PM_{10}$, which are measured by surface observation sites in the domains, is used

to consider the variability of aerosol properties.  Here, it is assumed that the mass of
aerosols that act as CCN is represented by $PM_{2.5}$ and $PM_{10}$ for the Seoul and Beijing areas,
respectively. The distance between the observation sites is ~ 1 km and the time interval



between observations of aerosol mass is ~ 10 minutes. Hence, the variability is represented
by fine spatiotemporal resolutions of the sites. The ground sites that are equipped with the
aerosol robotic network (AERONET; Holben et al., 2001) are in the domains. Distances
between these sites are ~10 km. In this study, $PM_{2.5}$/$PM_{10}$ data are used to represent the
spatiotemporal variability of aerosols acting as CCN over the domains and the simulation
periods. To represent aerosol composition and size distributions, data from the AERONET
sites are employed.

For the period with the observed clouds, based on the AERONET data, it is assumed

that on average, aerosol particles are internally mixed with 70 (80) % ammonium sulfate
and 30 (20) % organic compound for the Seoul (Beijing) case. This mixture is assumed to
represent aerosol chemical composition in the whole domain and during the entire
simulation period, based on the fact that aerosol composition does not vary significantly
over the domain and during the whole period with the observed clouds. Aerosols reflect,
scatter and absorb shortwave and longwave radiation before they are activated. This type
of aerosol-radiation interactions is not taken into account in this study. This is mainly based
on the fact that in the mixture, there is insignificant amount of radiation absorbers; black
carbon is a representative radiation absorbers. The AERONET observation indicates that
the size distribution of background aerosols acting as CCN follows the tri-modal log-
normal distribution for the Seoul (Beijing) case as exemplified in Figure 3a (3b). Hence, it
is assumed that for the whole domain and simulation period, the size distribution of
background aerosols acting as CCN follows the shape of distribution with specific size
distribution parameters (i.e., modal radius and standard deviation of each of nuclei,
accumulation and coarse modes, and the partition of aerosol number among those modes)
as shown in Figure 3a (3b) for the Seoul (Beijing) case. The assumed shape of the size
distribution of background aerosols in Figure 3a (3b) is with the average of the distribution
parameters over the AERONET sites and the period with clouds for the Seoul (Beijing)
case. Since the AERONET observation shows that the shape of the size distribution does
not vary significantly over the domain and during the simulation period for each of the
Seoul and Beijing cases, we believe that the assumption is reasonable. By using $PM_{2.5}$ and
$PM_{10}$ and based on the assumption of aerosol composition and size distribution above, the
background number concentrations of aerosols acting as CCN are obtained. These



background number concentrations, associated aerosol size distribution and composition
are interpolated or extrapolated to grid points immediately above the surface and time steps
in the simulation for each of the cases. There is no variation with height in background
aerosol concentrations from immediately above the surface to the top of the planetary
boundary layer (PBL). However, it is assumed that they decrease exponentially with height
from the PBL top upward. Aerosol size distribution and composition do not vary with
height. Once background aerosol properties (i.e., aerosol number concentrations, size
distribution and composition) are put into each grid point and time step, those properties at
each grid point and time step do not change during the course of the simulations.
For the control-s and control-b runs, aerosol properties of ice-nucleating particles
(INP) are not different from those of CCN except for the fact that the concentration of
background aerosols acting as CCN is 100 times higher than the concentration of
background aerosols acting as INP at each time step and grid point, following a general
difference between CCN and INP in terms of their concentrations (Pruppacher and Klett,

1978).

Once clouds form and background aerosols start to be in clouds, those aerosols are
not background aerosols anymore and the size distribution and concentrations of those
aerosols begin to evolve through aerosol sinks and sources that include advection and
aerosol activation (Fan et al., 2009). For example, once aerosols are activated, they are
removed from the corresponding bins of the aerosol spectra. In clouds, after aerosol
activation, aerosol mass starts to be inside hydrometeors and via collision-collection, it
transfers to different types and sizes of hydrometeors. In the end, aerosol mass disappears
in the atmosphere when hydrometeors with aerosol mass touches the surface. In non-cloudy
areas, aerosol size and spatial distributions are designed to be identical to the size and
spatial distributions of background aerosols, respectively. In other words, for this study,
we use "the aerosol recovery method". In this method, at any grid points, immediately after
clouds disappear entirely, aerosol size distributions and number concentrations recover to
background properties that background aerosols at those points have before those points
are included in clouds. In this way, we can keep aerosol concentrations outside clouds in
the simulations at observed counterparts. Thus, we are able to simulate aerosol evolutions,
via processes such as transportation of background aerosols by wind (or aerosol advection),


as observed, in case we neglect possible errors from the assumption on aerosol size
distribution and composition, and the process where observed data are interpolated or
extrapolated to grid points and time steps in the simulations. In the aerosol recovery method,
there is no time interval between the cloud disappearance and the aerosol recovery. Here,
when the sum of mass of all types of hydrometeors (i.e., water drops, ice crystals, snow
aggregates, graupel and hail) is not zero at a grid point, that grid point is considered to be
in clouds. When this sum becomes zero, clouds are considered to disappear. Many studies
using CSRM have employed this aerosol recovery method. They have proven that with the
recovery method, reasonable simulations of overall cloud and precipitation properties are
accomplished (e.g., Morrison and Grabowski, 2011; Lebo and Morrison, 2014; Lee et al.,
2016; Lee et al., 2018).

**3.3 Additional runs**


We repeat the control-s run by getting rid of aerosol-advection induced increases in aerosol
concentrations as a way of investigating how the aerosol advection affects the cloud system
in the Seoul area. This repeated run is named the low-aerosol-s run. An aerosol layer, which
is advected from East Asia or from the west of the Seoul area to it, increases aerosol
concentrations in the Seoul area. There are stations in islands in the Yellow Sea that
monitor the aerosol advection (Eun et al., 2016; Ha et al., 2019). To monitor and identify
the aerosol advection, $PM_{10}$ and $PM_{2.5}$ which are measured by a station in Baekryongdo
island in Yellow Sea are compared to those which are measured in stations in and around
the Seoul area. In Figure 2a, a dot and a rectangle mark the island and the Seoul area,
respectively. The time evolution of $PM_{2.5}$ measured by the station on the island and the
average $PM_{2.5}$ over stations in the Seoul area, between 07:00 LST on December 22[nd] and
21:00 LST on December 24[th] in 2017 when there is the strong advection of aerosols from
East Asia to the Seoul area, is shown in Figure 4. At 09:00 LST on December 22[nd], the
advection of aerosols from East Asia enables aerosol mass to start going up and attain its
peak around 05:00 LST on December 23[rd] on the island. Following this, aerosol mass starts
to increase in the Seoul area around 01:00 LST on December 23[rd], and the mass attains its



peak at 15:00 LST on December 23$^{rd}$ in the Seoul area. This is because aerosols, which are
advected from East Asia, move through the island to reach the Seoul area.
In the low-aerosol-s run, as a way of getting rid of the increase in aerosol
concentrations, it is assumed that PM2.5 and thus background aerosol concentrations after
01:00 LST on December 23$^{rd}$ do not evolve with the aerosol advection in the Seoul area.
Hence, the background concentration of aerosols acting as CCN is assumed to have that at
01:00 LST on December 23$^{rd}$ at each time step and grid point at the beginning of the
simulation period. However, background aerosol concentration acting as INP at each time
step and grid point in the low-aerosol-s run is not different from that in the control-s run
during the simulation period. In the observed PM data for the Seoul area, there is reduction
in PM by a factor of ~10 on average over a period between ~07:00 and ~14:00 LST on
December 24$^{th}$, since precipitation scavenges aerosols (Figure 4). To emulate this
scavenging and reflect it in background aerosols for the low-aerosol-s run, PM2.5 and
corresponding background aerosol concentrations at each grid point is gradually reduced
for the period between 07:00 and 14:00 LST on December 24$^{th}$. This reduction is done in
a way that background aerosol concentration at each grid point at 14:00 LST on December
24$^{th}$ is 10 times lower than that at 07:00 LST on December 24$^{th}$ in the low-aerosol-s run.
Then, PM2.5 and corresponding background aerosol concentrations at each grid point at
14:00 LST on December 24$^{th}$ maintains until the end of the simulation period. This results
in the evolution of the average PM2.5 over the Seoul area in the low-aerosol-s run as shown
in Figure 4. Here, the concentration of background aerosols acting as CCN, which is
averaged over the whole domain and simulation period, in the control-s run is 3.1 times
higher than that in the low-aerosol-s run. Via comparisons between the runs, how the
increasing concentration of background aerosols acting as CCN due to the aerosol
advection has an impact on clouds can be examined. The concentration of background
aerosols acting as CCN is different among grid points and time steps in the control-s run.
Hence, the ratio of the concentration of background aerosols acting as CCN between the
runs is different among gird points and time steps.
For the Beijing case, to examine how aerosols affect clouds and precipitation, we
repeat the control-b run with simply reduced concentrations of background aerosols acting
as CCN at each time step and grid point by a factor of 3.1. This repeated run is named the



304 low-aerosol-b run. The 3.1-fold increase in aerosol concentrations from the low-aerosol-b

305 run to the control-b is based on the 3.1-fold increase in the average concentration of

306 background aerosols acting as CCN from the low-aerosol-s run to the control-s run.

307 However, as in the control-s and low-aerosol-s runs, background aerosol concentration

308 acting as INP at each time step and grid point in the low-aerosol-b run is identical to that

309 in the control-b run during the simulation period. Hence, on average, a pair of the control-

310 s and low-aerosol-s runs has the same aerosol perturbation as in a pair of the control-b and

311 low-aerosol-b runs. Here, we define aerosol perturbation as a relative increase in aerosol

312 concentration when compared to that before the increase occurs. The brief summary of all

313 simulations in this study is given in Table 1.

315 **4. Results**

317 **4.1 Cumulative precipitation**

319 The automatic weather system (AWS) operates at the surface and rain gauges in the AWS

320 measure precipitation hourly with a spatial resolution of ~1 km. We compare the observed

321 precipitation to the simulated counterpart in the control-s run for the Seoul case and in the

322 control-b run for the Beijing case. For this comparison, the observed and simulated

323 precipitation rates at the surface are averaged over the domain for each of the Seoul and

324 Beijing cases (Figures 5a and 5b). Here, the simulated precipitation rates are smoothed

325 over 1 hour.  The comparison shows that the evolution of the simulated precipitation rate

326 does not deviate from the observed counterpart significantly (Figures 5a and 5b).

327  In the Seoul case, overall, the precipitation rate is higher in the control-s run than in

328 the low-aerosol-s run. As a result of this, the domain-averaged cumulative precipitation

329 amount at the last time step is 14.1 mm and 12.0 mm in the control-s run and the low-

330 aerosol-s run, respectively. The control-s run shows ~20 % higher cumulative precipitation

331 amount. In the Beijing case, the evolution of the mean precipitation rate in the control-b

332 run is not significantly different from that in the low-aerosol-b run. Due to this, the control-

333 b run shows only ~2% higher cumulative precipitation amount, despite the fact that the

334 concentrations of background aerosols are ~3 times higher in the control-b run than in the



low-aerosol-b run. Note that in the Seoul case, the time- and domain-averaged
concentration of background aerosols is also ~3 times higher in the control-s run than in
the low-aerosol-s run. Despite this, the difference in the cumulative precipitation amount
between the runs with different concentrations of background aerosols is greater in the
Seoul case than in the Beijing case.

**4.2 Precipitation, and associated latent-heat and dynamic processes**

Figures 6a and 6b show the cumulative frequency distributions of precipitation rates at the
last time step in the simulations for the Seoul and Beijing cases, respectively. In each of
those figures, the observed frequency distribution is shown and compared to the simulated
distribution. The observed distribution is obtained by interpolating and extrapolating the
observed precipitation rates to grid points and time steps in each of the control-s and
control-b runs. The observed maximum precipitation rates are 13.0 and 44.5 mm hr$^{-1}$ for
the Seoul and Beijing cases, respectively, and these maximum rates are similar to those in
the control-s and control-b runs, respectively. Overall, the observed and simulated
frequency distributions are in good agreement for each of the cases. This enables us to
assume that results in the control-s (control-b) run are benchmark results to which results
in the low-aerosol-s (low-aerosol-b) run can be compared to identify how aerosols have an
impact on clouds and precipitation for the Seoul (Beijing) case. Here, it is notable that for
the Beijing case, while differences in the cumulative precipitation amount between the
control-b and low-aerosol-b runs are not significant, features in the frequency distribution
of precipitation rates between those runs are substantially different (Figure 6b).

**1) Seoul case**

**a. Precipitation Frequency distributions**

Regarding precipitation whose rates are higher than ~2 mm hr$^{-1}$, the cumulative
precipitation frequency at the last time step is higher in the control-s run as compared to
that in the low-aerosol-s run (Figure 6a). There are increases in the cumulative frequency



by a factor of as much as ~10 in the control-s run when it comes to specific ranges of
precipitation whose rates are higher than ~2 mm hr$^{-1}$. When it comes to precipitation rates
above ~11 mm hr$^{-1}$, precipitation is present in the control-s run and  precipitation is absent
in the low-aerosol-s run. Regarding precipitation whose rates are lower than ~2 mm hr$^{-1}$,
differences in the cumulative frequency between the runs are insignificant. Hence, we see
that there are significant increases in the frequency of relatively heavy precipitation whose
rates are above ~2 mm hr$^{-1}$ in the control-s run when compared to that in the low-aerosol-
s run. At the last time step, this results in a larger amount of cumulative precipitation in the
control-s run than in the low-aerosol-s run.
The time evolution of the cumulative precipitation frequency is shown in Figure 7. At
06:00 LST December 24$^{th}$ 2017, which corresponds to the initial stage of the precipitation
development, the maximum precipitation rate reaches ~3 mm hr$^{-1}$ and there is the greater
frequency over most of precipitation rates in the control-s run than in the low-aerosol-s run.
With the time progress from 06:00 to 10:00 LST, the maximum precipitation rate increases
to reach 12 mm hr$^{-1}$ and the cumulative frequency is higher over precipitation whose rates
are higher than ~3 mm hr$^{-1}$ in the control-s run, while for precipitation whose rates are
lower than ~3 mm hr$^{-1}$, differences in the cumulative frequency between the runs are
negligible (Figures 7a, 7b, 7c and 7d). When time reaches 12:00 LST, which is around a
time when the peak in the evolution of the area-averaged precipitation rates occurs and thus
the system is at its mature stage, the maximum precipitation rate increases up to ~13 mm
hr$^{-1}$. The basic patterns of differences in the cumulative precipitation frequency between
the runs with the maximum precipitation rate around 13 mm hr$^{-1}$, which are established at
12:00 LST, maintain until the end of the simulation period (Figures 6a and 7e).

**b.  Condensation, deposition, updrafts and associated variables**

Note that the source of precipitation is precipitable hydrometeors which are raindrops,
snow, graupel and hail particles. Droplets and ice crystals are the source of those
precipitable hydrometeors mostly via collision and coalescence among droplets and ice
crystals. Droplets and ice crystals gain their mass via condensation and deposition,
respectively. Based on this, to explain the greater cumulative precipitation amount in the


control-s run than in the low-aerosol-s run, the evolutions of differences in condensation,
deposition and associated updrafts between the runs are analyzed.  The vertical profiles of
differences in the area-averaged condensation and deposition rates, updraft mass fluxes and
associated variables between the runs at 03:20, 03:40, 06:00 and 12:00 LST are shown in
Figure 8. Condensation rates in the control-s run start to be larger than that in the low-
aerosol-s run throughout all altitudes at 03:20 LST (Figure 8a). Higher aerosol
concentrations induce more nucleation of droplets and associated greater integrated surface
of droplets. Hence, more droplet surface is provided for water vapor to condense onto.  This
induces more water vapor to be condensed to lead to more condensation in the control-s
run. This establishes stronger feedbacks between updrafts and condensation, leading to
greater droplet (or cloud-liquid) mass (Figure 8a).  Then, these stronger feedbacks, which
involve stronger updrafts particularly above 2 km in altitude, subsequently induce greater
deposition and snow mass at 03:40 LST in the control-s run.
Through aerosol-induced stronger feedbacks between condensation, deposition and
updrafts in the control-s run, differences in condensation and deposition between the
control-s and low-aerosol-s runs increase as time progresses from 03:40 LST to 06:00 LST
(Figures 8b and 8c). At 06:00 LST, there is more freezing starting to occur in the control-
s run than in the low-aerosol-s run. However, differences in freezing are two to three orders
of magnitude smaller than those in condensation and deposition. A similar situation in
terms of relative differences between freezing, condensation and deposition continues after
06:00 LST until time reaches 12:00 LST when the overall differences in the cumulative
precipitation frequency between the runs are established (Figures 8c and 8d). It is notable
that at 12:00 LST, the freezing rate becomes lower in the control-s run. Hence, here, we
see that aerosol-induced more cumulative precipitation amount and associated differences
in the precipitation frequency distribution between the control-s and low-aerosol-s runs are
primarily associated with aerosol-induced more condensation which induce aerosol-
induced more deposition but weakly connected to aerosol-induced changes in freezing.

**c. Condensation frequency distributions and horizontal distributions of**
**condensation and precipitation**





Based on the importance of condensation for aerosol-induced changes in precipitation, the
horizontal distribution of the column-averaged condensation rates over the domain and the
cumulative frequency distribution of the column-averaged condensation rates at each time
step is obtained. To better visualize the role of condensation in precipitation, the horizontal
distribution of the column-averaged condensation rates is superimposed on that of
precipitation rates (Figure 9). At 03:40 LST, condensation mainly occurs around the
northern part of the domain as marked by a yellow rectangle. The synoptic wind condition
in the marked area favors the collision between northward and southward wind and the
associated convergence around the surface (Figures 9a and 9b). This convergence induces
updrafts and condensation in the marked area.  In the marked area, more aerosols induce
more and more extensive condensation, which leads to the higher domain-averaged
condensation rates in the control-s run than in the low-aerosol-s run (Figures 8b, 9a and
9b). More droplets are formed on more aerosols and more droplets provide more surface
areas where condensation occurs and this enables more and more extensive condensation
in the control-s run than in the low-aerosol-s run (Figures 8b, 9a and 9b). At 06:20 LST, a
precipitating system is advected into the domain via the western boundary (Figures 9c and
9d). As seen in Figures 9c, 9d, 9e and 9f for 06:20 and 07:20 LST, respectively, as time
progresses from 06:20 to 07:20 LST, the advected precipitating system is further advected
to the east and extended mostly over areas in the northern part of the domain where
condensation mainly occurs. This confirms that condensation is the main source of cloud
mass and precipitation. In the eastern part of the domain, there are mountains and in
particular, higher mountains are on the northeastern part of the domain than in the other
parts of the domain. These higher mountains induce forced convection and associated
condensation more effectively in the northeastern part than in the other parts. This is in
favor of the precipitating system that extends further to the east in the northern part of the
domain. Due to more aerosols, condensation, which is induced by forced convection over
mountains, is more and more extensive in the control-s run (Figures 9c, 9d, 9e and 9f).
As time progresses to 08:40 LST, the precipitating system moves eastward further in
the northern part of the domain and the system in the control-s run extends to the east
further as compared to that in the low-aerosol-s run (Figures 9g, 9h, 9i, 9j, 9k and 9l). In
association with more aerosols and associated more condensation over mountains in the



northeastern part, there is more extension of the system in the control-s run than in the low-
aerosol-s run. This enables the system in the control-s run to reach the eastern boundary at
08:40 LST, which is earlier than in the low-aerosol-s run. The system in the low-aerosol-s
run reaches the eastern boundary at 09:00 LST (Figure 9n). Here, we see that although
aerosols do not change overall locations of the precipitation system, they affect how fast
the system extends to the east by affecting the amount of condensation which is produced
by forced convection. Associated with this, as seen in Figure 10, the control-s run has the
much higher cumulative condensation frequency than the low-aerosol-s run over all of
condensation rates during the period between 07:20 and 09:00 LST. Contributed by this,
the higher precipitation frequency over most of precipitation rates occurs in the control-s
run during the period (Figures 7b and 7c).
At 10:00 LST, in the southern part of the domain, there is a precipitating area forming
as marked by a yellow rectangle (Figures 9o and 9p). The precipitation area in the southern
part of the domain extends and merge into the advecting main precipitating system in the
northern part of the domain during the period between 10:20 and 11:00 LST (Figures 9q,
9r, 9s and 9t). The merge leads to precipitation that occupies most of the domain at 12:00
LST (Figures 9u and 9v). After 10:00 LST, associated with this merge, the maximum
precipitation rate increases to 13 mm hr$^{-1}$ at 12:00 LST (Figures 7e). After 13:00 LST, the
precipitation enters its dissipating stage and its area reduces and nearly disappears during
the period between 13:00 and 15:20 LST (Figures 9w, 9x, 9y and 9z). Even after the merge,
aerosol-induced more condensation maintains and this in turn contributes to a situation
where the control-s run has the greater precipitation frequency over most of precipitation
rates than in the low-aerosol-s run until the simulations progress to their last time step
(Figures 6a, 7e and 8d).

**2) Beijing case**

Stronger convection and deeper clouds develop in the Beijing case than in the Seoul case.
The maximum cloud depth is ~ 7 and 12 km in the control-s and control-b runs, respectively.
In the Seoul case, clouds do not reach the tropopause, while they reach the tropopause in
the Beijing case. Deeper clouds in the Beijing case produce the maximum precipitation rate



of ~45 mm hr⁻¹ in the control-b run. However, less deep clouds in the Seoul case produce
the maximum precipitation rate of ~13 mm hr⁻¹ in the control-s run (Figure 6).

**a.   Precipitation frequency distributions**


When it comes to precipitation whose rates are higher than ~12 mm hr⁻¹, the control-b run
has the higher cumulative precipitation frequency at the last time step than the low-aerosol-
b run (Figure 6b). The cumulative frequency increases by a factor of as much as ~10 with
respect to some ranges of precipitation rates above ~12 mm hr⁻¹. Moreover, regarding
precipitation rates higher than ~33 mm hr⁻¹, precipitation is present in the control-b run,
however, precipitation is absent in the low-aerosol-b run. Hence, we see that the frequency
of comparatively heavy precipitation whose  rates are higher than ~12 mm hr⁻¹ rises
significantly in the control-b run as compared to that in the low-aerosol-b run. Below ~2
mm hr⁻¹, there is also the greater precipitation frequency in the control-b run than in the
low-aerosol-b run. Unlike the situation for precipitation rates above ~12 mm hr⁻¹ and below
~2 mm hr⁻¹, for precipitation rates from ~2 to ~12 mm hr⁻¹, the control-aerosol-b run has
the lower precipitation frequency than in the low-aerosol-b run. Here, we see that the higher
precipitation frequency above ~12 mm hr⁻¹ and below ~2 mm hr⁻¹ balances out the lower
precipitation frequency between ~2 and ~12 mm hr⁻¹ in the control-b run. This results in
the similar cumulative precipitation amount between the runs.

Figure 11 shows the time evolution of the cumulative precipitation frequency. When

precipitation starts around 16:00 LST, the higher precipitation frequency occurs over most
of precipitation rates in the low-aerosol-run-b run than in the control-b run (Figure 11a).
At 16:00 LST, the maximum precipitation rate is lower than 1.0 mm hr⁻¹ for both of the
runs. As the time progresses to 17:00 LST, the maximum precipitation rate increases to
~17 mm hr⁻¹, and the cumulative precipitation frequency starts to be higher (lower) over
precipitation rates higher than ~12 mm hr⁻¹ (between ~2 and ~12 mm hr⁻¹) in the control-b
run than in the low-aerosol-b run (Figure 11b). At 17:20 LST, the frequency starts to be
greater when it comes to precipitation rates below 2 mm hr⁻¹ together with those above ~
12 mm hr⁻¹ in the control-b run, while the lower frequency between 2 and 12 mm hr⁻¹ in
the control-b run maintains (Figure 11c). At 17:20 LST, the maximum precipitation rate



increases to 42 (19) mm hr$^{-1}$ in the control-b (low-aerosol-b) run (Figure 11c). At 19:00
LST, the maximum precipitation rate increases to ~45 (33) mm hr$^{-1}$ for the control-b (low-
aerosol-b) run, while the qualitative nature of differences in the precipitation frequency
distributions with the tipping precipitation rates of ~2 and ~12 mm hr$^{-1}$ between the runs
does not vary much between 17:20 and 19:00 LST (Figures 11c and 11d). The qualitative
nature of differences in the cumulative precipitation frequency between the runs and the
maximum precipitation rates in each of the runs, which are established at 19:00 LST, do
not vary significantly until the end of the simulation period (Figures 6b and 11d).

**b. Condensation, deposition, updrafts and associated variables**


As done for the Seoul case, as a way of better understanding differences in the cumulative
precipitation amount and frequency between the control-b and low-aerosol-b runs, the
evolutions of differences in the vertical distributions of the area-averaged condensation
rates, deposition rates, freezing rates, cloud-liquid and snow mass density and updrafts
mass fluxes are obtained and shown in Figures 12. As seen in Figure 5b, precipitation starts
around 16:00 LST but differences in condensation rates start at 14:20 LST with higher
condensation rates in the control-b run (Figure 12a). Similar to the situation in the Seoul
case, higher aerosol concentrations induce more nucleation of droplets and associated
greater integrated surface of droplets. Hence, more droplet surface is provided for water
vapor to condense onto. This induces more water vapor to be condensed to leads to more
condensation in the control-b run. Due to this, cloud liquid as a source of precipitation
becomes greater in the control-b run (Figure 12a). Increased condensation rates induce
increased condensational heating and thus intensified updrafts (Figure 12a). When time
reaches 15:40 LST, deposition rates start to show differences between the runs, however,
unlike the situation in the Seoul case, higher aerosol concentrations results in lower
deposition rates in the control-b run (Figure 12b). When time reaches 16:00 LST,
differences in freezing start to occur and freezing rates are lower in the control-b run
(Figure 12c). However, due to stronger updrafts, which are mainly ascribed to more
condensation, deposition rates start to be higher in the control-b run at a portion of altitudes
with non-zero differences in deposition rates between the runs (Figure 12c). At 17:20 LST,



due to stronger updrafts again, which are in turn mainly due to more condensation, freezing
rates also start to be higher in the control-b run at a portion of altitudes with non-zero
differences in freezing rates between the runs (Figure 12d). As the time progresses to 19:00
LST, deposition rates are lower in the control-b run over most of the altitudes with non-
zero differences in deposition rates between the runs and freezing rates are lower in the
control-b run over all of altitudes with non-zero differences in freezing rates between the
runs (Figure 12e). As the time progresses to 19:00 LST, higher condensation rates maintain
in the control-b run over most of the altitudes with non-zero differences in condensation
rates between the runs (Figure 12e). Here, differences in freezing rates are two to three
orders of magnitude smaller than those in condensation and deposition between the runs.
Associated with this, the time- and domain-averaged deposition and freezing (condensation)
rates are lower (higher) in the control-b run over the whole simulation period, although the
average differences in freezing rates are negligible as compared to those in deposition rates
between the runs. Hence, more condensation (but not deposition and freezing) is a main
cause of stronger updrafts in the control-b run. Remember that condensation (deposition
and freezing) acts as a source of droplets (ice crystals) which are in turn a source of
precipitable hydrometeors and precipitation through collision and coalescence processes.
Thus, more condensation tends to induce increases in the precipitation rate in the control-
b run.  Less deposition and freezing tend to induce decreases in precipitation rate in the
control-b run.  This competition between condensation, deposition and freezing leads to
negligible differences in the cumulative precipitation amount at the last time step between
the control-b and low-aerosol-b runs, although roles of freezing in this competition are
negligible as compared to those of condensation and deposition.

**c.  Condensation   frequency   distributions,   horizontal   distributions   of**

**condensation   and   precipitation,   and   condensation-precipitation**

**correlations**


Figure 13 shows the horizontal distribution of the column-averaged condensation rates over
the domain and Figure 14 shows the cumulative frequency distributions of column-
averaged condensation rates at selected times. As in the Seoul case, the horizontal





distribution of condensation rates is superimposed on that of precipitation rates and the
terrain in Figure 13. At 14:20 LST, condensation starts to occur in places with mountains,
which induce forced convection, and condensation is concentrated around the center of the
domain as marked by a yellow circle (Figures 13a and 13b). Note that condensation does
not occur in the plain area which is the south of the 100-m terrain-height contour line
(Figures 13a and 13b).  Due to higher aerosol concentrations, there is more condensation
around the center in the control-b run than in the low-aerosol-b run (Figures 13a and 13b).
This leads to a situation where the control-b run has the higher area-averaged condensation
rates than the low-aerosol-b run (Figure 12a). Then, as time progresses to 17:20 LST, the
condensation area extends to the eastern and western parts of the domain mostly over
mountain areas (Figures 13c and 13d). Hence, the main source of condensation is
considered to be forced convection over mountains. As seen in Figures 13c and 13d, higher
aerosol concentrations induce the control-b run to have much more condensation spots and
thus much bigger areas with condensation than the low-aerosol-b run at 17:20 LST.
Associated with this, aerosol-induced more condensation in the control-b run maintains
with the time progress to 17:20 LST (Figure 12d).  At 17:20 LST, precipitation mainly
occurs in a spot which is in the western part of areas with relatively high condensation rates
(Figures 13c and 13d).

At 17:20 LST, as seen in the cumulative frequency of condensation rates, the control-

b run has the higher condensation frequency above condensation rate of $\sim10 \times 10^{-6}$ g m$^{-3}$ s$^{-}$
$^1$ and below that of $\sim3\times10^{-6}$ g m$^{-3}$ s$^{-1}$ than the low-aerosol-b run (Figure 14a). This pattern
of differences in the condensation frequency distribution with the tipping condensation-
rate points at $\sim10 \times 10^{-6}$ and $\sim3 \times 10^{-6}$ g m$^{-2}$ s$^{-1}$ continues up to 19:00 LST (Figures 14b).
Figure 15 shows the mean precipitation rate over each of the column-averaged
condensation rates for the period up to 17:20 LST in the control-b run. A column-averaged
condensation rate in an air column with a precipitation rate at its surface is obtained and
these condensation and precipitation rates are paired at each column and time step. Then,
collected precipitation rates are classified and grouped based on the corresponding paired
column-averaged condensation rates. The classified precipitation rates corresponding to
each of the column-averaged condensation rates are averaged arithmetically to construct
Figure 15. There are only less than 10% differences in the mean precipitation rate for each





of the column-averaged condensation rates between the control-b and low-aerosol-b runs
(not shown). Figure 15 shows that generally a higher condensation rate is related to a higher
mean precipitation rate. It is also roughly shown that, according to the mean precipitation
rate for each condensation rate, overall, condensation rates below $\sim3 \times 10^{-6}$ g m$^{-3}$ s$^{-1}$ and
above $\sim10 \times 10^{-6}$ g m$^{-3}$ s$^{-1}$ are correlated with precipitation rates below $\sim2$ mm hr$^{-1}$ and
above $\sim12$ mm hr$^{-1}$, respectively, while condensation rates between $\sim3$ and $\sim10 \times 10^{-6}$ g
m$^{-3}$ s$^{-1}$ are correlated with precipitation rates between $\sim2$ and $\sim12$ mm hr$^{-1}$ (Figure 15).
Hence, on average, the higher frequency of condensation with rates above $\sim10 \times 10^{-6}$ g m$^{-3}$
$^{-3}$ s$^{-1}$ and below $\sim 3 \times 10^{-6}$ g m$^{-3}$ s$^{-1}$ can be considered to lead to the higher frequency of
precipitation whose rates are higher than $\sim 12$ mm hr$^{-1}$ and lower than $\sim 2$ mm hr$^{-1}$ in the
control-b run, respectively. It can also be considered that the lower condensation frequency
between $\sim3$ and $\sim10 \times 10^{-6}$ g m$^{-3}$ s$^{-1}$ leads to the lower precipitation frequency between $\sim2$
and $\sim12$ mm hr$^{-1}$ in the control-b run.  At 17:20 LST, the larger precipitation frequency
between $\sim2$ and $\sim12$ mm hr$^{-1}$ in the low-aerosol-b run nearly offsets the larger precipitation
frequency in the other ranges of precipitation rates in the control-b run (Figure 11c). This
leads to the similar average precipitation rate between the runs at 17:20 LST and
contributes to the similar cumulative precipitation at the last time step between the runs
(Figure 5b).

**d.  Evaporation and gust fronts**

As time progresses from 17:00 to 19:00 LST, the precipitation system moves northward
(Figure 16). At the core of the precipitation system, due to evaporation and downdrafts,
there is the horizontal outflow forming at 17:00 LST (Figures 16a and 16b). The core is
represented by the field of precipitation whose rates are higher than 1 mm hr$^{-1}$ in Figure 16.
At the core, the northward outflow is magnified by the northward synoptic-scale wind,
while at the core, the outflow in the other directions is offset by the northward synoptic-
scale wind. Hence, the outflow is mainly northward from 17:00 LST onwards as marked
by yellow circles in Figures 16. This enables convergence or a gust front, which is produced
by the outflow from the core, to be mainly formed at the north of the core. Note that the
intensity of a gust front is proportional to that of outflow from a core of precipitation or



convective system (Weisman and Klemp, 1982; Houze, 1993). The strong gust front at the
north of the core generates strong updrafts, a significant amount of condensation and
precipitation. Then, a subsequent area with clouds and precipitation is formed at the north
of the core as time progresses, which means that the precipitation system extends or moves
to the north as seen in comparisons between sub-panels with different times in Figure 16.
This movement, which is induced by collaborative work between outflow, synoptic wind
and gust fronts, is typical in deep convective clouds.

During the period between 17:00 and 19:00 LST, the time- and domain-averaged

cloud-liquid or droplet evaporation rate is higher and downdrafts are stronger, although the
control-b run has the lower average rain evaporation rate than the low-aerosol-b run (Figure
17). This is consistent with the numerous previous studies that have shown more
evaporation of droplets and associated stronger downdrafts with higher aerosol
concentrations (e.g., Tao et al., 2007; Tao et al., 2012; Khain et al., 2008; Lee et al., 2018).
Due to higher condensation rates and lower autoconversion rates, the control-b run has
more cloud liquid or droplets as a source of evaporation and this enables more evaporative
cooling and stronger downdrafts in the control-b run. For the period from 17:00 to 19:00
LST, with the development of convergence or the gust front, as mentioned above, the
maximum precipitation rate increases from $\sim$ 17 (17) to $\sim$ 45 (33) mm hr$^{-1}$ in the control-b
(low-aerosol-b) run (Figure 11). This indicates that the gust-front development contributes
to the overall intensification of the precipitation system, while it moves northward. If there
were only northward synoptic-scale wind with no formation of the gust front, the system
would move northward with less intensification.  Over the period from 17:00 to 19:00 LST,
stronger downdrafts and associated stronger outflow generate a stronger gust front and
more subsequent condensation, leading to a situation where the control-b run has the higher
maximum precipitation rate than the low-aerosol-b run. Around 19:00 LST, the system
enters its dissipating stage, accompanying reduction in the precipitating area and the area-
averaged precipitation rate (Figures 5b, 16m and 16n).

**e.  Moist static energy**




Analyses to construct Figure 15 are repeated only for a time point at 16:30 LST and they
indicate that as shown in Figure 15 for the period up to 17:20 LST, the column-averaged
condensation rates above $10 \times 10^{-6}$ g m$^{-3}$ s$^{-1}$ and below $3 \times 10^{-6}$ g m$^{-3}$ s$^{-1}$ correspond to
precipitation rates above 12 mm hr$^{-1}$ and below 2 mm hr$^{-1}$, respectively, at 16:30 LST.
According to those analyses, condensation rates between $3 \times 10^{-6}$ g m$^{-3}$ s$^{-1}$ and $10 \times 10^{-6}$ g
m$^{-3}$ s$^{-1}$ correspond to precipitation rates between 2 and 12 mm hr$^{-1}$ at 16:30 LST.
Condensation, which controls droplet mass and precipitation, is controlled by updrafts and
updrafts are in turn controlled by instability. One of important factors that maintain
instability is the moist static energy. Motivated by this, to better understand differences in
the precipitation frequency distribution in association with those in the condensation
frequency distribution between the control-b and low-aerosol-b runs, we calculate the flux
of the moist static energy and the flux is defined as follows:

$$\vec{Fs} = S \times \rho \times \vec{V} \qquad\qquad (1),$$

where $\vec{Fs}$ represents the flux of the moist static energy, S the moist static energy, $\rho$ the air
density and $\vec{V}$ the horizontal-wind vector. In Eq. (1), we see that the flux is in the vector
form and has two components, which are its magnitude and direction.  The fluxes of the
moist static energy in the PBL are obtained over the domain at 16:30 LST, since in general,
the moist static energy in the PBL has much stronger effects on instability and updrafts
than that above the PBL. In particular, as diagrammatically depicted in Figure 18, we focus
on the PBL fluxes of the energy that cross the boundary over a time step at 16:30 LST
between areas with the column-averaged condensation rate from $3 \times 10^{-6}$ g m$^{-3}$ s$^{-1}$ to 10
$\times 10^{-6}$ g m$^{-3}$ s$^{-1}$, which are referred to as "area A", and those with the column-averaged
condensation rate above $10 \times 10^{-6}$ g m$^{-3}$ s$^{-1}$, which are referred to as "area B". This is because
we are interested in the exchange of the moist static energy between areas A and B and this
exchange can be seen by looking at those fluxes which cross the boundary between those
areas.

We are interested in the exchange of the energy, since we hypothesized that the

exchange somehow alters instability in each of areas A and B in a way that there are
increases (decreases) in instability, the updraft intensity, condensation and precipitation





with increasing aerosol concentrations in area B (A), leading to the higher (lower)
frequency of condensation whose rates are higher than $10 \times 10^{-6}$ g m$^{-3}$ s$^{-1}$ (between $3\times10^{-6}$
g m$^{-3}$ s$^{-1}$ and $10 \times 10^{-6}$ g m$^{-3}$ s$^{-1}$ ) and precipitation whose rates are higher than 12 mm hr$^{-1}$
(between 2 and 12 mm hr$^{-1}$) in the control-b run than in the low-aerosol-b run. When the
PBL fluxes, which crosses the boundary over the time step at 16:30 LST, are averaged over
the domain at 16:30 LST, there is the net flux from area A to area B. This means that there
is the net transportation of the moist static energy from areas with condensation rates
between $3\times10^{-6}$ g m$^{-3}$ s$^{-1}$ and $10 \times 10^{-6}$ g m$^{-3}$ s$^{-1}$ to those with condensation rates greater
than $10 \times 10^{-6}$ g m$^{-3}$ s$^{-1}$ in the PBL at 16:30 LST as shown in Table 2. Table 2 shows the
average flux of the moist static energy which crosses the boundary between areas A and B
in the control-b run as well as  the low-aerosol-b run. To calculate the average flux at 16:30
LST in Table 2, the fluxes, which cross the boundary between areas A and B over the time
step at 16:30 LST, only at grid points in the PBL are summed and divided by the number
of all grid points over the whole domain at 16:30 LST. Here, all grid points include grid
points with those fluxes crossing the boundary and those without those fluxes including
grid points with zero fluxes, and include grid points both in and above the PBL. For the
calculation, the flux from area A to area B has a positive sign, while the flux from area B
to area A has a negative sign as depicted in Figure 18. Since the average flux is positive
for both of the runs, there is on average, the net flux from area A to area B in the PBL. The
above-described analysis for the average of the fluxes crossing the boundary between areas
A and B is repeated for every time step between 16:30 and 17:00 LST and based on this,
the average flux over the period between 16:30 and 17:00 LST is obtained. Analyses to
construct Figure 15 are repeated again for a time period between 16:30 and 17:00 LST.
These repeated analyses find that as in the situation only for 16:30 LST, the column-
averaged condensation rates below $3 \times 10^{-6}$ g m$^{-3}$ s$^{-1}$  and above $10 \times 10^{-6}$ g m$^{-3}$ s$^{-1}$
correspond to precipitation rates below 2 mm hr$^{-1}$ and above 12 mm hr$^{-1}$, respectively,
while the column-averaged condensation rates between $3\times10^{-6}$ g m$^{-3}$ s$^{-1}$ and $10 \times 10^{-6}$ g m$^{-3}$ s$^{-1}$
$^3$ s$^{-1}$ correspond to precipitation rates between 2 and 12 mm hr$^{-1}$ for every time step between
16:30 and 17:00 LST. As shown in Table 2, the average flux for the period between 16:30
and 17:00 LST is also positive as in the situation only for 16:30 LST. This means on



average, there is the net transportation of the moist static energy from area A to area B in
the PBL during the period between 16:30 and 17:00 LST.

At 16:30 LST, condensation with rates above $10 \times 10^{-6}$ g m$^{-3}$ s$^{-1}$ starts to develop and

this forms area B. Area B has stronger updrafts via greater condensational heating than in
other areas, including area A, with lower condensation rates. Stronger updrafts in area B
induce the convergence of air and associated moist static energy from area A to area B.
Since the average condensation rate and updrafts at 16:30 LST over area B are higher and
stronger due to increasing aerosol concentrations, respectively, the air convergence and the
associated transportation of the moist static energy in the PBL from area A to area B are
stronger and more, respectively, in the control-b run than in the low-aerosol-b run (Table
2). Stated differently, area B steals the moist static energy from area A, and this occurs
more effectively in the control-b run. This increases instability and further intensifies
updrafts in area B, and decreases instability and weakens updrafts in area A, while these
increases and decreases (intensification and weakening) of instability (updrafts) are greater
in the control-b run for the period from 16:30 to 17:00 LST. This increases condensation,
cloud mass and precipitation whose rates are higher than 12 mm hr$^{-1}$ in area B, and
decreases condensation, cloud mass and precipitation whose rates are from 2 to 12 mm hr$^{-}$
$^{1}$ in area A. These increases and decreases occur more effectively for the control-b run than
for the low-aerosol-b run during the period. This in turn leads to the lower precipitation
frequency for the precipitation rates from 2 to 12 mm hr$^{-1}$ and the higher frequency for the
precipitation whose rates are higher than  12 mm hr$^{-1}$ at 17:00 LST in the control-b run
(Figure 11b). The weakened updrafts and reduced condensation turn a portion of
precipitation with rates between 2 and 12 mm hr$^{-1}$ to precipitation whose rates are below 2
mm hr$^{-1}$, and this takes place more efficiently in the control-b run during the period between
16:30 and 17:00 LST. This eventually increases the frequency of precipitation rates below
2 mm hr$^{-1}$ and this increase is greater for the control-b run, leading to the greater
precipitation frequency for the precipitation rates below 2 mm hr$^{-1}$ in the control-b run at
17:20 LST (Figure 11c).

**5.  Discussion**




### 5.1 Comparison of the Seoul and Beijing cases



In this section, we compare the Seoul case to the Beijing case. To examine aerosol effects
on the Beijing case, the control-b and the low-aerosol-b runs are compared, and remember
that the low-aerosol-b run is the repeated control-b run by simply reducing the background-
aerosol concentrations at each of grid points and time steps by a factor of 3.1. Remember
that the low-aerosol-s run is the repeated control-s run and the low-aerosol-s run has the
3.1 times lower concentration of background aerosols, which is averaged over the whole
domain and simulation period, than the control-s run. Hence, the relative variation of the
average concentration of background aerosols from the control-b run to the low-aerosol-b
run is not different from that from the control-s run to the low-aerosol-s run. This enables
the two pairs of runs to be compared for an identical magnitude of an aerosol perturbation.
Note that clouds in the Seoul case are less deep as compared to those in the Beijing
case. Thus, overall, clouds and associated updrafts in the Seoul case are not as strong as
those in the Beijing case. Hence, unlike the situation in the Beijing case, stronger updrafts,
which accompany higher condensation rates, and associated convergence in the Seoul case
are not strong enough to steal the sufficient amount of the moist static energy from weaker
updrafts which accompany lower condensation rates. This makes the redistribution of the
moist static energy between areas with relatively higher condensation rates and those with
relative lower condensation rates, such as that between areas A and B for the Beijing case,
ineffective for the Seoul case. Due to this, the sign of aerosol-induced changes in the
frequency of precipitation rates does not vary throughout all of the precipitation rates
except for the range of low precipitation rates where there are nearly no aerosol-induced
changes in the frequency in the Seoul case as shown in Figure 6a. As seen in Figure 6a,
mainly due to increases in condensation and deposition, precipitation frequency increases
for most of precipitation rates, although the precipitation frequency does not show
significant changes as aerosol concentration increases for relatively low precipitation rates
in the control-s run as compared to that in the low-aerosol-s run. This means that there are
no tipping precipitation rates where the sign of aerosol-induced changes in the frequency
of precipitation rates changes in the Seoul case, contributing to the higher cumulative



precipitation amount in the simulation with higher aerosol concentrations for the Seoul
case, which are different from the situation in the Beijing case.
In the Beijing case with deeper clouds as compared to those in the Seoul case, clouds
develop gust fronts via strong downdrafts and associated strong outflow. These gust fronts
play an important role in developing strong convection and associated high precipitation
rates. Unlike the situation in the Seoul case, there are strong clouds and associated updraft
entities that are able to steal heat and moisture (or the moist static energy) as sources of
instability from areas with relatively less strong clouds and updrafts with medium strength.
This further intensifies strong clouds and weakens clouds with medium strength. Due to
this, the cumulative frequency of heavy (medium) precipitation in association with strong
clouds (clouds with medium strength) increases (decreases). Some of the weakened clouds
eventually produce light precipitation, which increase the cumulative frequency for light
precipitation. The intensification of strong clouds and the weakening of clouds with
medium strength gets more effective with increasing aerosol concentrations. Hence, in the
Beijing case, for medium precipitation in association with clouds with medium strength,
the simulation with higher aerosol concentrations shows the lower cumulative precipitation
frequency at the last time step. However, for heavy precipitation, which is associated with
strong clouds, and light precipitation, the simulation with higher aerosol concentrations
shows the higher cumulative precipitation frequency at the last time step. These differential
responses of precipitation to increasing aerosol concentrations among different types of
precipitation occur in the circumstances of the similar cumulative precipitation amount
between the simulations with different aerosol concentrations. This similar precipitation
amount is due to above-mentioned competition between aerosol-induced changes in
condensation, deposition and freezing.
In both of the Seoul and Beijing cases, aerosol-induced changes in condensation plays
an important role in making differences in the precipitation amount and/or the precipitation
frequency distribution between the simulations with different aerosol concentrations. It is
notable that in less deep clouds in the Seoul case, in addition to condensation, deposition
plays a role in precipitation to induce aerosol-induced increases in the precipitation amount.
Aerosol-induced increases in condensation initiate the differences in cloud mass and
precipitation and then aerosol-induced increases in deposition follow to further enhance



those differences. However, in deep clouds in the Beijing case, condensation tends to
induce increases in cloud mass and precipitation with increasing aerosol concentrations,
while deposition tend to induce decreases in cloud mass and precipitation with increasing
aerosol concentrations. Hence, as clouds get shallower and thus ice processes become less
active, the role of ice processes in aerosol-induced changes in precipitation amount turns
from aerosol-induced suppression of precipitation to enhancement of precipitation. Here,
we find that contrary to the traditional understanding, the role of aerosol-induced variation
of freezing in precipitation is not significant as compared to that of condensation and
deposition in both of the cases.
**6.  Summary and conclusions**
This study examines impacts of aerosols on clouds and precipitation in two metropolitan
areas, which are the Seoul and Beijing areas, in East Asia that has experienced substantial
increases in aerosol concentrations over the last decades. The examination is performed via
simulations, which use a CSRM. These simulations are for deep clouds which reach the
tropopause in the Beijing case and for comparatively less deep clouds which do not reach
the tropopause yet grow above the level of freezing in the Seoul case.
In both of the cases, aerosol-induced changes in condensation plays a critical role in
aerosol-induced variation of precipitation properties (e.g., the precipitation amount and the
precipitation frequency distribution). In the Seoul case, aerosol-induced increases in
condensation and subsequent increases in deposition lead to aerosol-induced increases in
the precipitation frequency over most of precipitation rates and thus in the precipitation
amount; note that condensation and deposition are sources of cloud mass and precipitation.
However, in the Beijing case, while there are increases in condensation with increasing
aerosol concentrations, there are decreases in deposition with increasing aerosol
concentrations. So, there is competition between increases in condensation and decreases
in deposition. This competition leads to negligible aerosol-induced changes in cumulative
precipitation amount in the Beijing case. Also, there is another competition for the moist
static energy among clouds with different updrafts and condensation in the Beijing case.



This competition results in aerosol-induced differential changes in the precipitation
frequency distributions.
With clouds getting deeper from the Seoul case to the Beijing case, clouds and
associated updrafts, which are strong enough to steal the moist static energy from other
clouds and their updrafts, appear. This makes strong clouds stronger and clouds with
medium strength weaker. With higher aerosol concentrations, strong clouds steal the more
energy, and thus strong clouds become stronger and clouds with medium strength weaker
with a greater magnitude. As a result of this, there are more frequent heavy precipitation
(whose rates are higher than 12 mm hr$^{-1}$) and light precipitation (whose rates are lower than
2 mm hr$^{-1}$), and less frequent medium precipitation (with rates from 2 to 12 mm hr$^{-1}$) with
increasing aerosol concentrations in the Beijing case.
In both of the Seoul and Beijing cases, there are mountains and they play an important
role in how cloud and precipitation evolve with time and space. In both of the cases, the
precipitating system moves or expands over mountains which induce forced convection
and generate condensation. This important role of mountains and forced convection in the
formation and evolution of the precipitation system has not been examined much in the
previous studies of aerosol-cloud interactions, since many of those previous studies (e.g.,
Jiang et al., 2006; Khain et al., 2008; Li et al., 2011; Morrison et al., 2011) have dealt with
convective clouds that develop over plains and oceans. Hence, findings in this study, which
are related to mountain-forced convection and its interactions with aerosols, can be
complementary to those previous studies. Stated differently, this study can shed light on
our path to the understanding of aerosol-cloud interactions over more general domains not
only with no terrain but also with terrain.










**Code/Data availability**

Our private computer system stores the code/data which are private and used in this study.
Upon approval from funding sources, the data will be opened to the public. Projects related
to this paper have not been finished, thus, the sources prevent the data from being open to
the public currently. However, if information on the data is needed, contact the
corresponding author Seoung Soo Lee (slee1247@umd.edu).
**Author contributions**
Essential initiative ideas are provided by SSL, KJH and KHS to start this work. Simulation
and observation data are analyzed by SSL, JC and GK. JU and YZ reviewed the results and
contributed to their improvement.
**Competing interests**
The authors declare that they have no conflict of interest.

**Acknowledgements**
This study is supported by the National Research Foundation of Korea (NRF) grant funded
by the Korea government (MSIT) (No. NRF2020R1A2C1003215), the "Construction of
Ocean Research Stations and their Application Studies" project funded by the Ministry of
Oceans and Fisheries, South Korea. Authors thank Danhong Dong at Chinese Academy of
Sciences and Fang Wu at Beijing Normal University for their reviewing this paper.














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



**FIGURE CAPTIONS**

Figure 1. Wind at 850 hPa level (m s$^{-1}$), equivalent potential temperature (K), and
geopotential height (m) over Northeast Asia at (a) 21:00 LST December 23$^{rd}$ 2017 and (b)
09:00 LST July 27$^{th}$ 2015. Wind, equivalent potential temperature, and geopotential height
are represented by arrows, a shaded field, and contours, respectively.  The rectangle in the
Korean Peninsula in (a) marks the Seoul area and that in the East-Asia continent in (b)
marks the Beijing area.
Figure 2. Rectangles in (a) and (b) mark the Seoul area in the Korean Peninsula and the
Beijing area in the East-Asia continent, respectively. A dot in (a) marks Baekryongdo
island. In (a) and (b), the light blue represents the ocean and the green the land area.

Figure 3. Surface size distribution of aerosols (a) for the Seoul case and (b) for the Beijing
case. Aerosol number concentration per unit volume of air is represented by N and aerosol
diameter by D.

Figure 4. Time series of PM$_{2.5}$ observed at the ground station in Baekryongdo island (blue
line) and of the average PM$_{2.5}$ over ground stations in the Seoul area (red line) between
07:00 LST on December 22$^{nd}$ and 21:00 LST on December 24$^{th}$ in 2017.  Note that PM$_{2.5}$
observed at stations in the Seoul area is applied to the control-s run whose period is marked
by the dashed rectangle. Time series of the average PM$_{2.5}$ over stations in the Seoul area in
the low-aerosol-s run for the simulation period is also shown (black solid line).

Figure 5. Time series of precipitation rates at the surface, which are averaged over the
domain and smoothed over 1 hour, (a) for the control-s and low-aerosol-s runs in the Seoul
area and (b) for the control-b and low-aerosol-b runs in the Beijing area. In (a) and (b), the
averaged and observed precipitation rates over the observation sites in the Seoul and
Beijing areas, respectively, are also shown.



Figure 6. Observed and simulated cumulative frequency distributions of precipitation rates
at the surface for (a) the Seoul case, which are collected over the Seoul area, and (b) the
Beijing case, which are collected over the Beijing area, at the last time step. Simulated
distributions are in the control-s and low-aerosol-s runs for the Seoul case and in the
control-b and low-aerosol-b runs for the Beijing case. The observed distribution is obtained
by interpolating and extrapolating the observed precipitation rates to grid points and time
steps in the control-s and control-b runs for the Seoul and Beijing cases, respectively.
Figure 7. Cumulative frequency distributions of the precipitation rates at the surface in the
control-s and low-aerosol-s runs for the Seoul case at (a) 06:00, (b) 07:20, (c) 09:00, (d)
10:00 and (e) 12:00 LST.
Figure 8. Vertical distributions of differences in the area-averaged condensation,
deposition and freezing rates, and cloud-liquid and snow mass density, and updraft mass
fluxes between the control-s and low-aerosol-s runs at (a) 03:20, (b) 03:40, (c) 06:00 and
(d) 12:00 LST.
Figure 9. Spatial distributions of terrain heights, column-averaged condensation rates and
precipitation rates at the surface at (a) and (b) 03:40, (c) and (d) 06:20, (e) and (f) 07:20,
(g) and (h) 08:00, (i) and (j) 08:20, (k) and (l) 08:40, (m) and (n) 09:00, (o) and (p) 10:00,
(q) and (r) 10:20, (s) and (t) 11:00, (u) and (v)12:00, (w) and (x) 13:00, (y) and (z) 15:20
LST. The distributions in the control-s run are shown in (a), (c), (e), (g), (i), (k), (m), (o),
(q), (s), (u), (w) and (y), and the distributions in the low-aerosol-s run are shown in (b), (d),
(f), (h), (j), (l), (n), (p), (r), (t), (v), (x) and (z). Condensation rates are shaded. Dark-yellow
and dark-red contours represent precipitation rates at 0.5 and 3.0 mm hr$^{-1}$, respectively,
while beige, light brown and brown contours represent terrain heights at 100, 300 and 600
m, respectively. See text for yellow rectangles in (a), (b), (o) and (p)
Figure 10. Cumulative frequency distributions of the column-averaged condensation rates
in the control-s and low-aerosol-s runs for the Seoul case at (a) 07:20 and (b) 09:00 LST.





Figure 11. Cumulative frequency distributions of the precipitation rates at the surface in
the control-b and low-aerosol-b runs for the Beijing case at (a) 16:00, (b) 17:00, (c) 17:20,
and (d) 19:00 LST.

Figure 12. Vertical distributions of differences in the area-averaged condensation,
deposition and freezing rates, and cloud-liquid and snow mass density, and updraft mass
fluxes between the control-b and low-aerosol-b runs at (a) 14:20, (b) 15:40, (c) 16:00, (d)
17:20 and (e) 19:00 LST.

Figure 13.  Spatial distributions of terrain heights, column-averaged condensation rates and
precipitation rates at the surface at (a) and (b) 14:20, and (c) and (d) 17:20 LST. (a) and (c)
are for the control-b run and (b) and (d) are for the low-aerosol-b run.  Condensation rates
are shaded. Dark-yellow and dark-red contours represent precipitation rates at 1.0 and 2.0
mm hr$^{-1}$, respectively, while beige, light brown, brown and dark brown contours represent
terrain heights at 100, 500, 1000 and 1500 m, respectively.  Yellow circles in (a) and (b)
mark places where most of condensation occurs.

Figure 14. Cumulative frequency distributions of the column-averaged condensation rates
in the control-b and low-aerosol-b runs for the Beijing case at (a) 17:20 and (b) 19:00 LST.

Figure 15. Mean precipitation rates corresponding to each column-averaged condensation
rate for the control-b run. One standard deviation of precipitation rates is represented by a
vertical bar at each condensation rate.

Figure 16. Spatial distributions of precipitation rates (shaded) and wind vectors (arrows)
for the Beijing case at (a) and (b) 17:00, (c) and (d) 17:20, (e) and (f) 17:40, (g) and (h)
18:00, (i) and (j) 18:20, (k) and (l) 18:40, and (m) and (n) 19:00 LST. The distributions in
the control-b run are in (a), (c), (e), (g), (i), (k) and (m). The distributions in the low-
aerosol-b run are in (b), (d), (f), (h), (j), (l) and (n).



Figure 17. Vertical distributions of (a) the area-averaged cloud-liquid and rain evaporation
rates and (b) downdraft mass fluxes in the control-b and low-aerosol-b runs over a period
from 17:00 to 19:00 LST.

Figure 18. Diagram that depicts the flux of the moist static energy (arrows) between area
A (blue) and area B (red). See text for details.


























| Simulations | Site | Concentrations of background aerosols acting as CCN |
|---|---|---|
| Control-s run | Seoul area | Observed and affected by the aerosol advection |
| Low-aerosol-s run | Seoul area | Same as those in the control-s run but unaffected by the aerosol advection |
| Control-b run | Beijing area | Observed |
| Low-aerosol-b run | Beijing area | Reduced by a factor of 3.1 as compared to those observed |


Table 1. Summary of simulations


| Simulations | The average flux of the moist static energy which crosses the boundary between areas A and B ($J\ m^{-2}\ s^{-1}$) | |
|---|---|---|
| | At 16:30 LST | 16:30 to 17:00 LST |
| Control-b run | $1.5\times10^5$ | $1.7\times10^5$ |
| Low-aerosol-b run | $1.1\times10^5$ | $1.2\times10^5$ |


Table 2. Average flux of the moist static energy which crosses the boundary between areas
A and B at 16:30 LST and for a period from 16:30 to 17:00 LST.







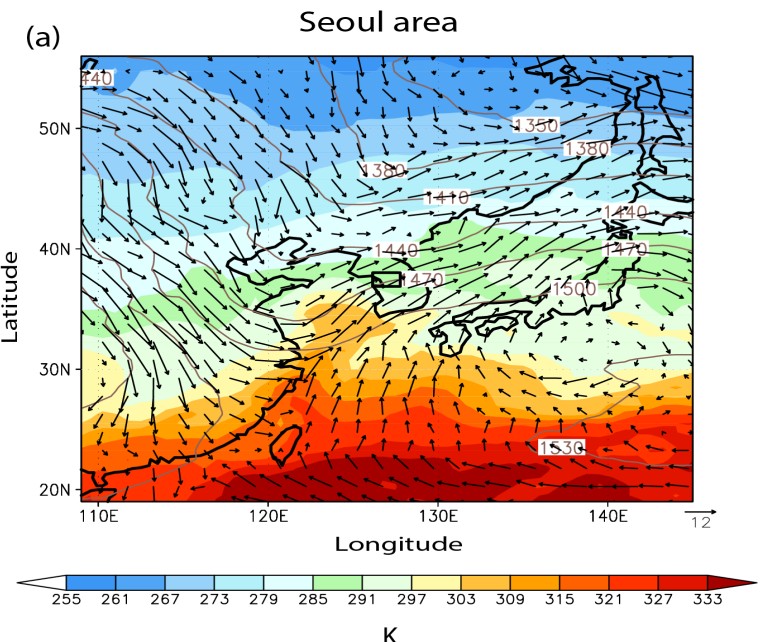

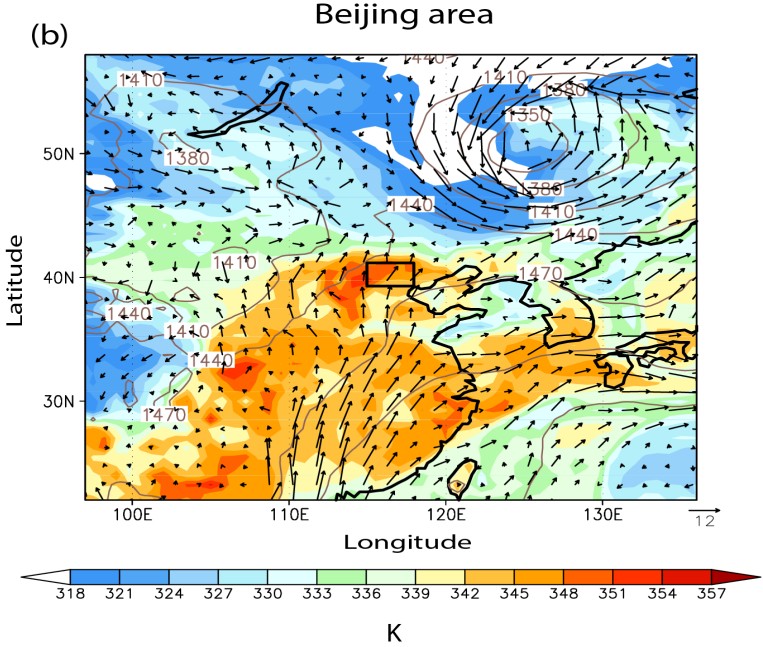


**Figure 1**




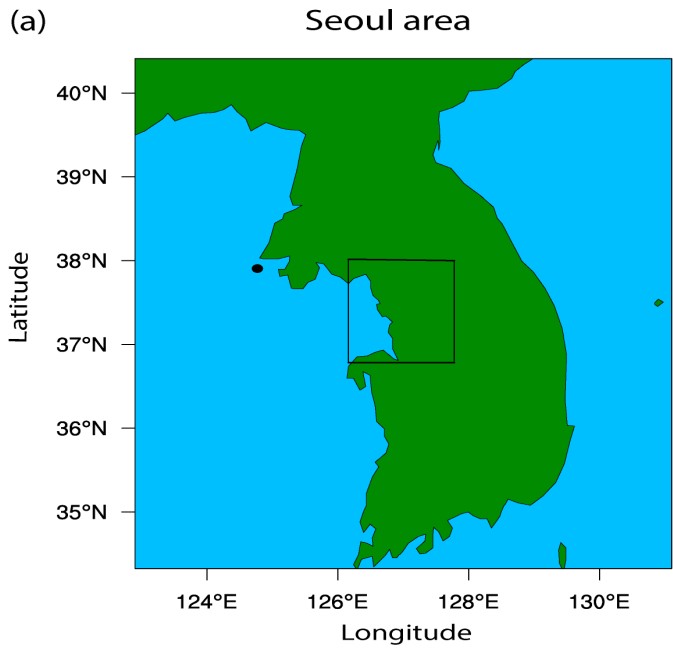

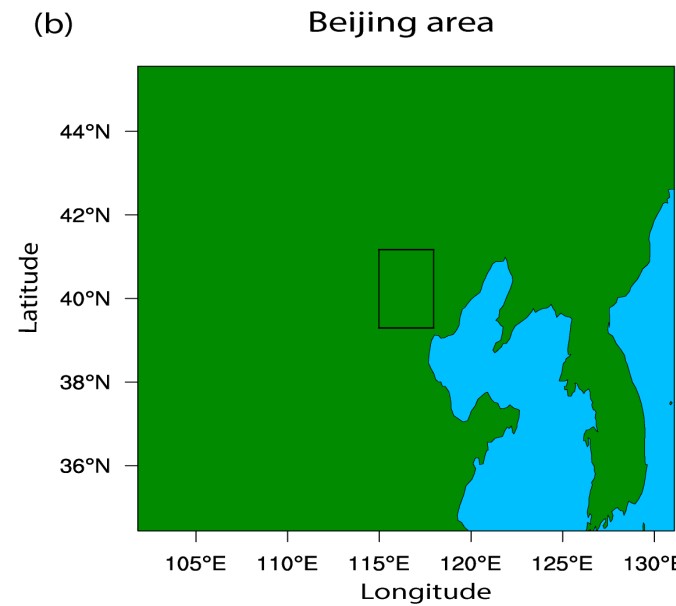


**Figure 2**



(a)

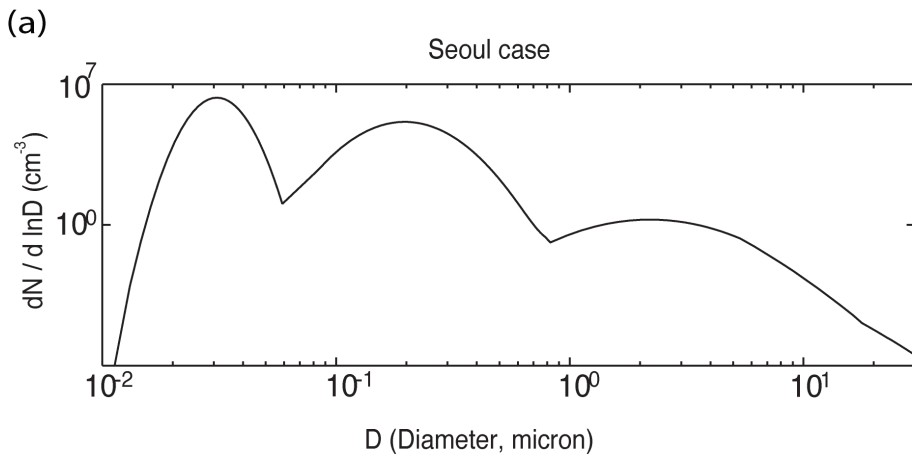

(b)

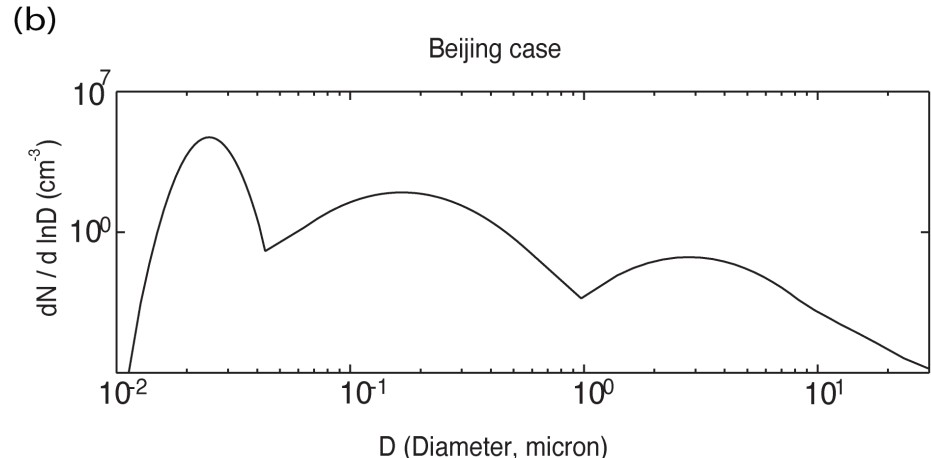


**Figure 3**










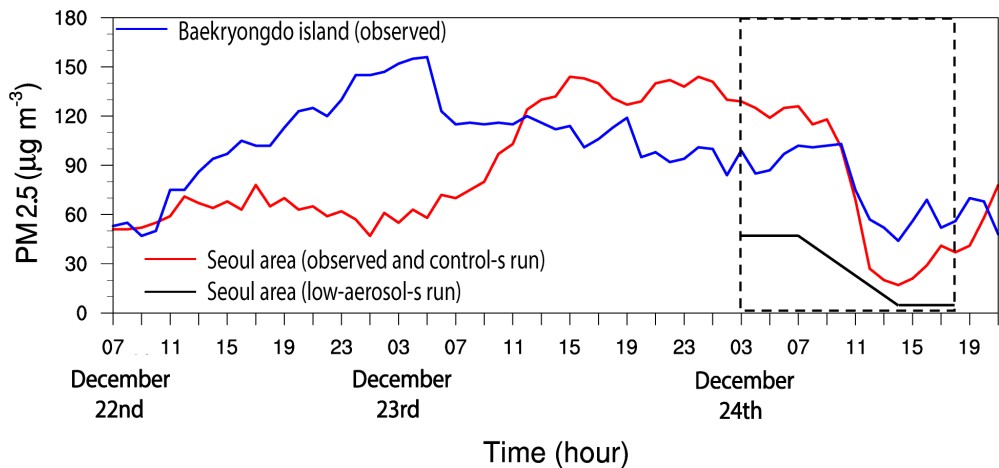


**Figure 4**
















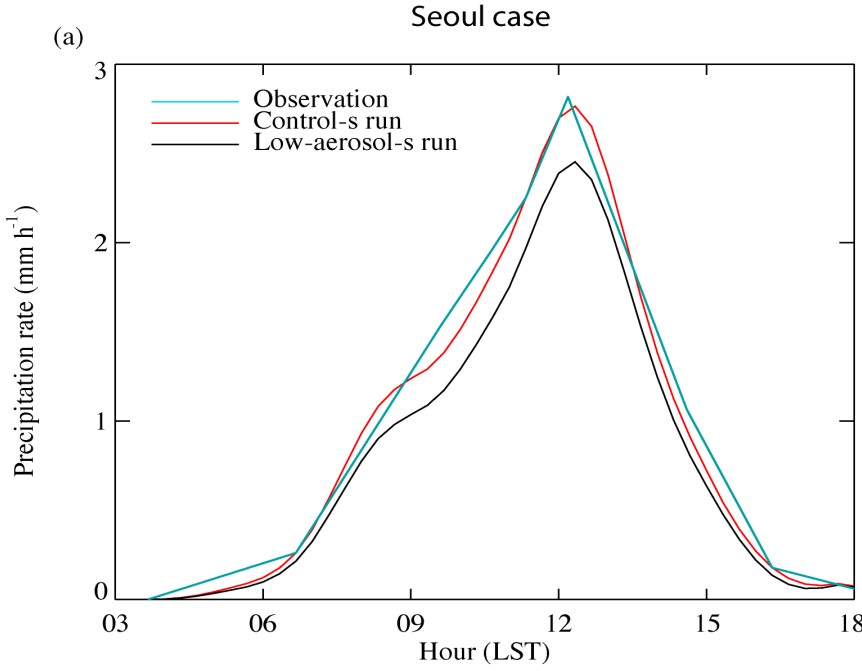

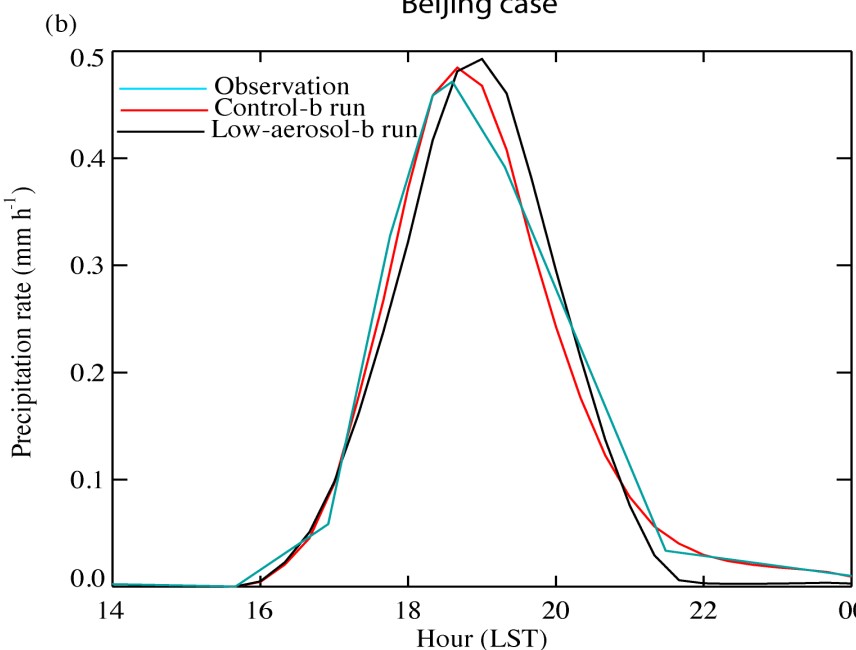


1214                                     **Figure 5**



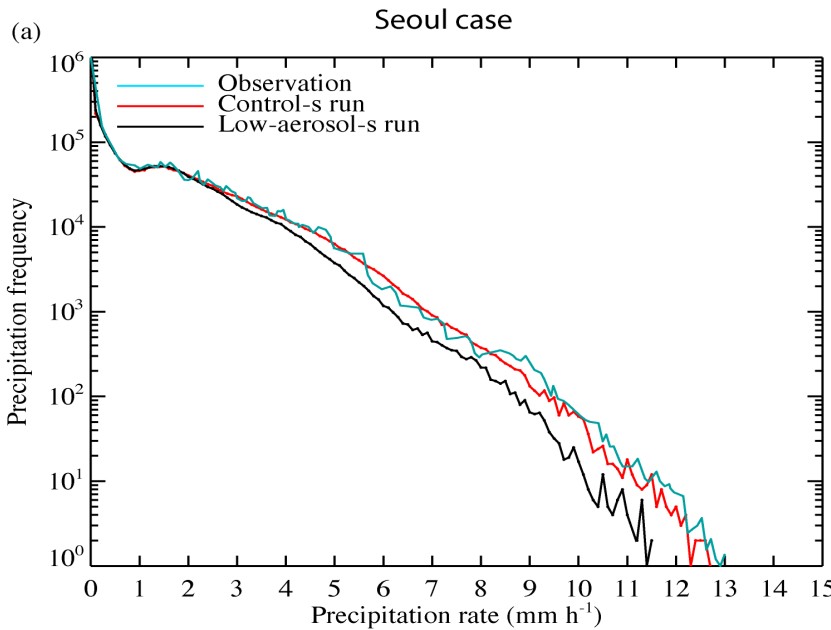

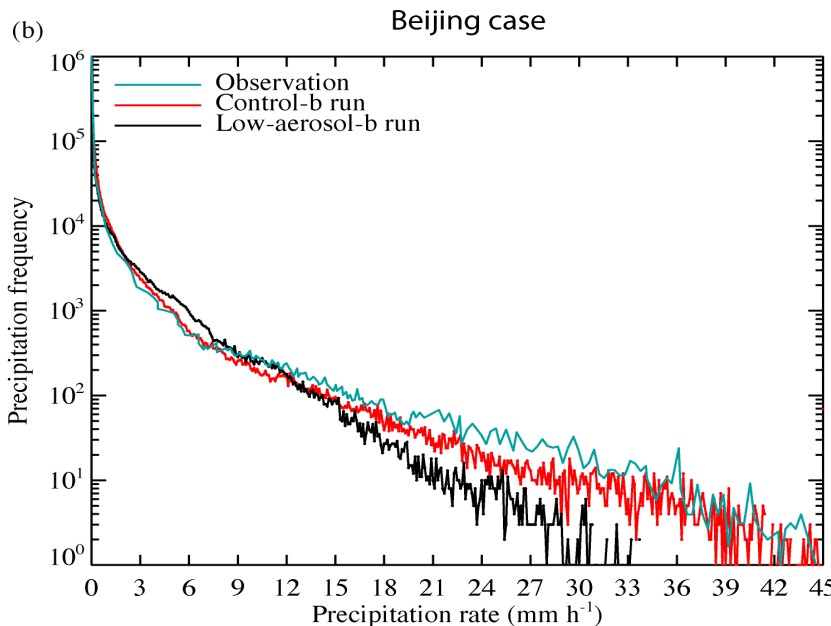


**Figure 6**






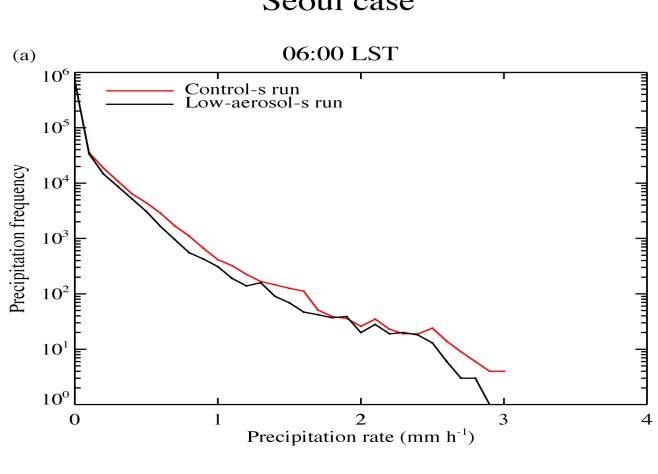

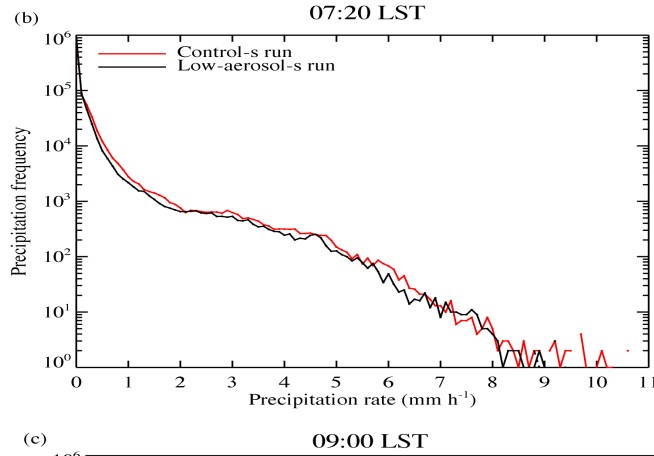

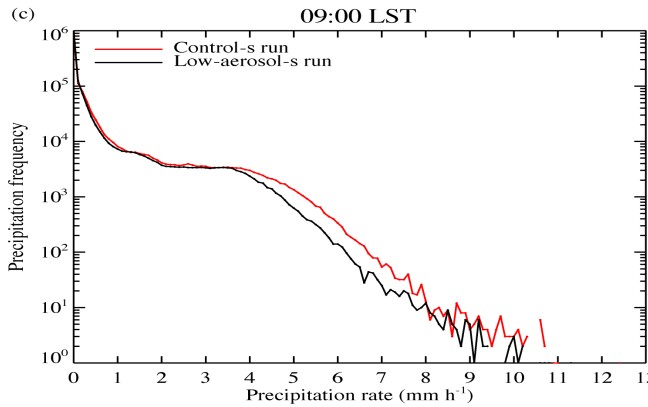


**Figure 7**






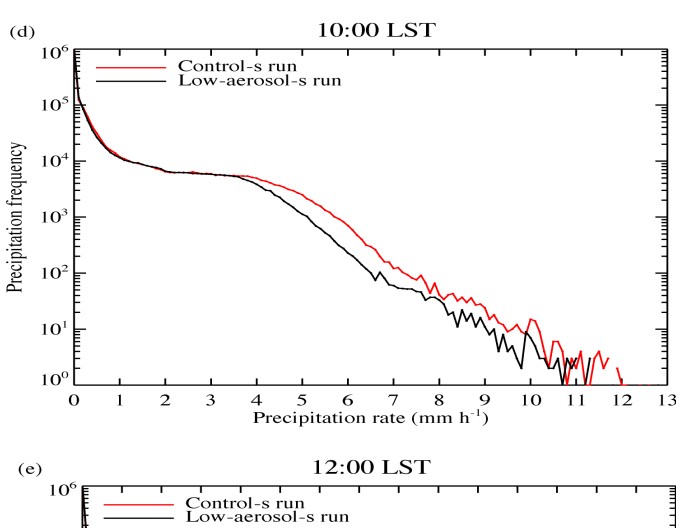

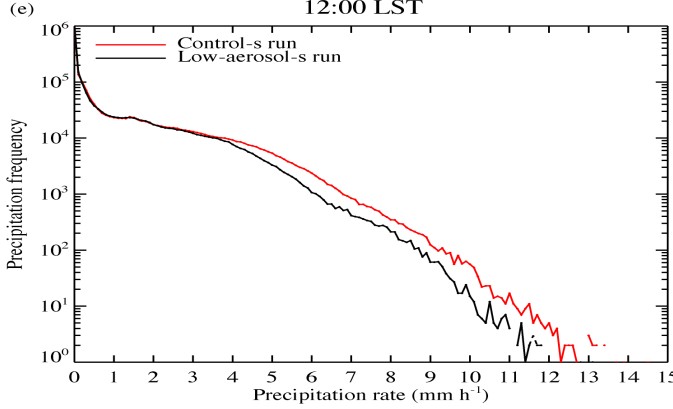


**Figure 7**












# Seoul case
# (control-s run minus low-aerosol-s run)

(a)                03:20 LST

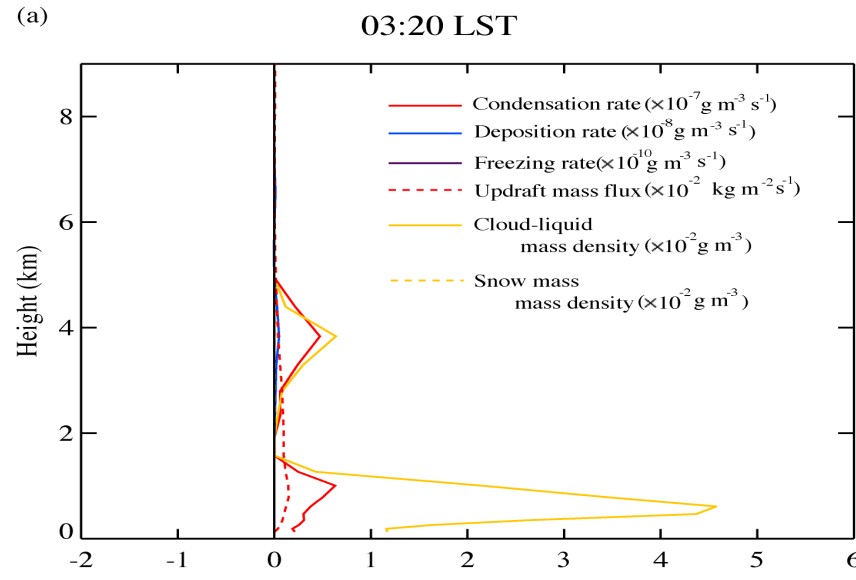

(b)                03:40 LST

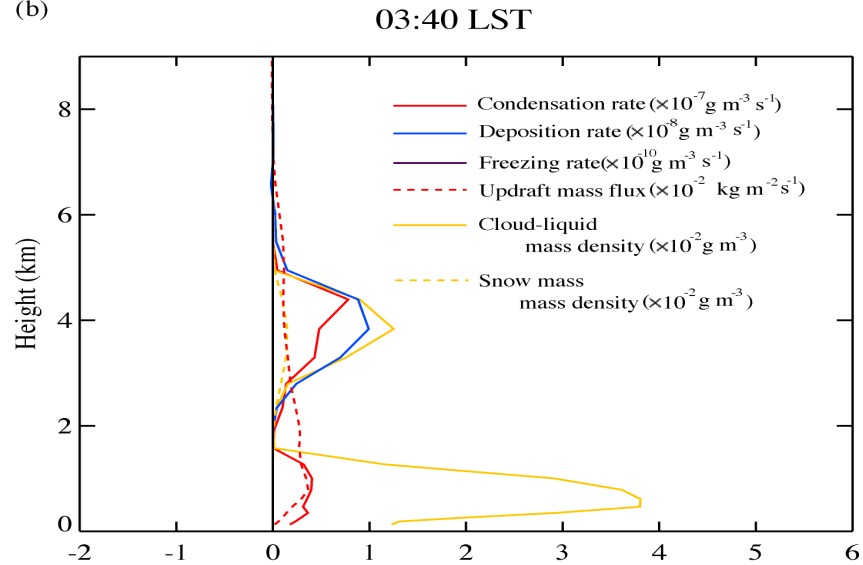


1228                **Figure 8**

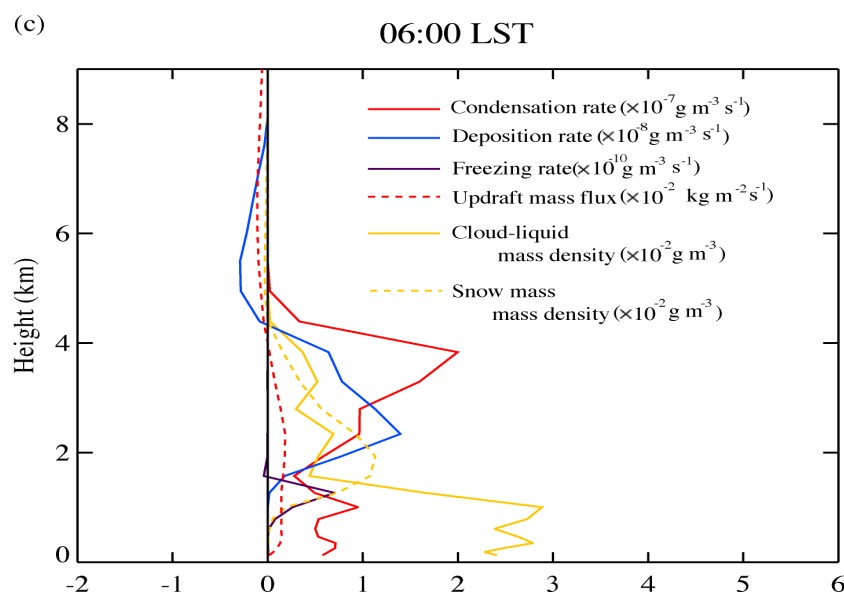

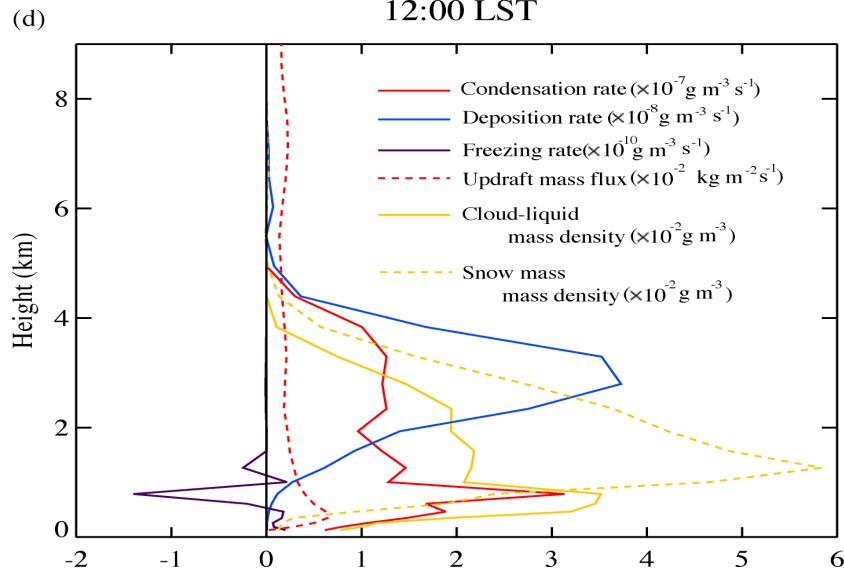


1230                    **Figure 8**




**Figure 9**











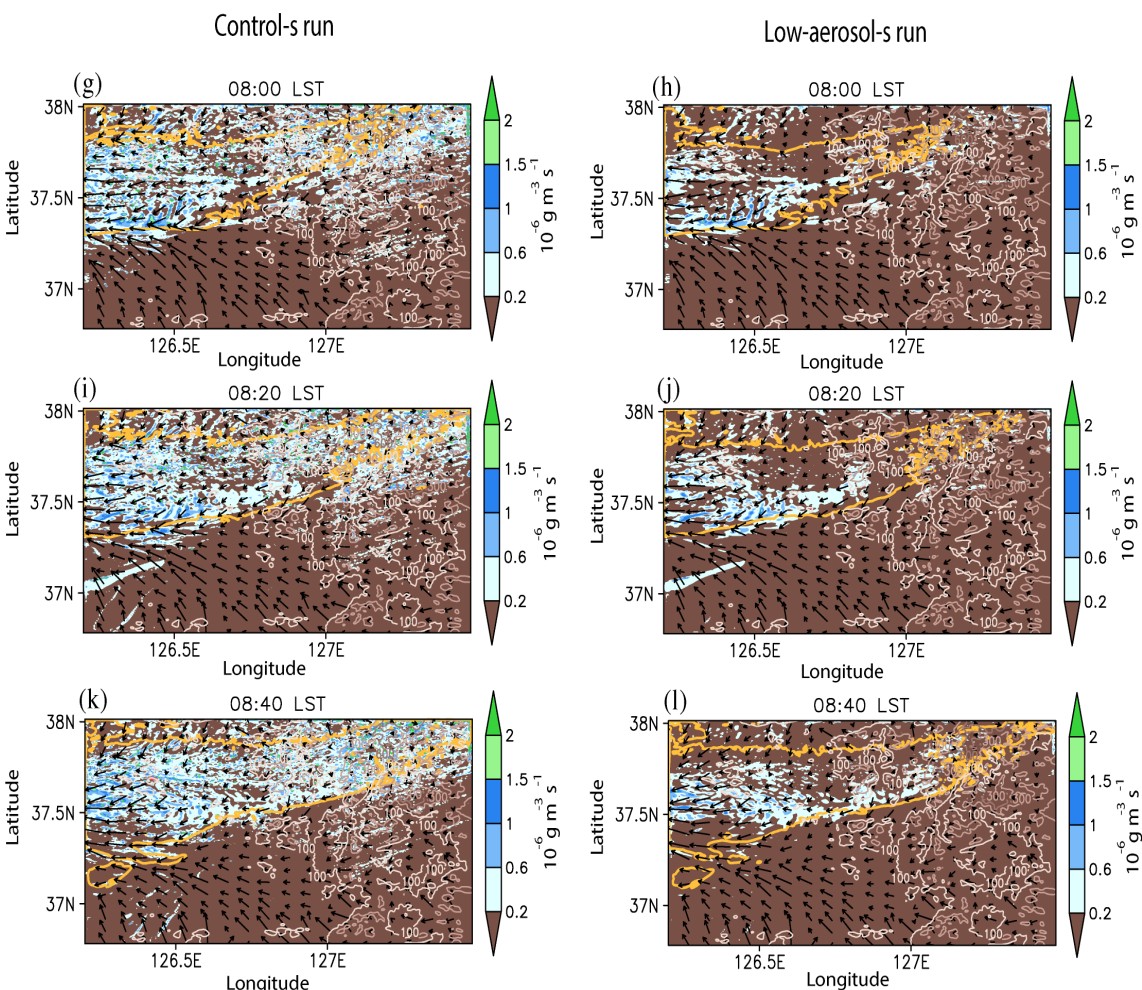

**Figure 9**





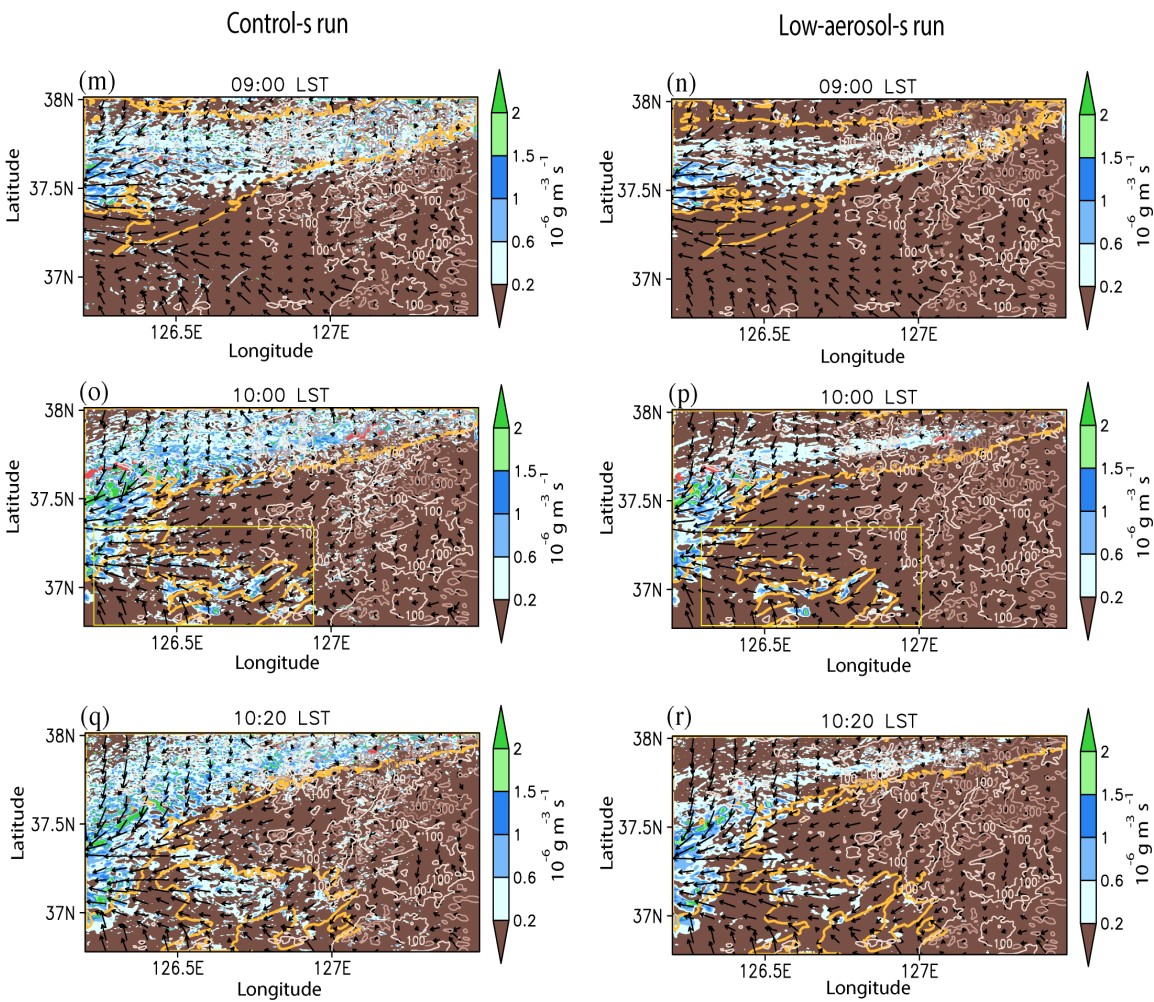


1245                                            **Figure 9**









Figure 9





## Seoul case

**(a)**  07:20 LST

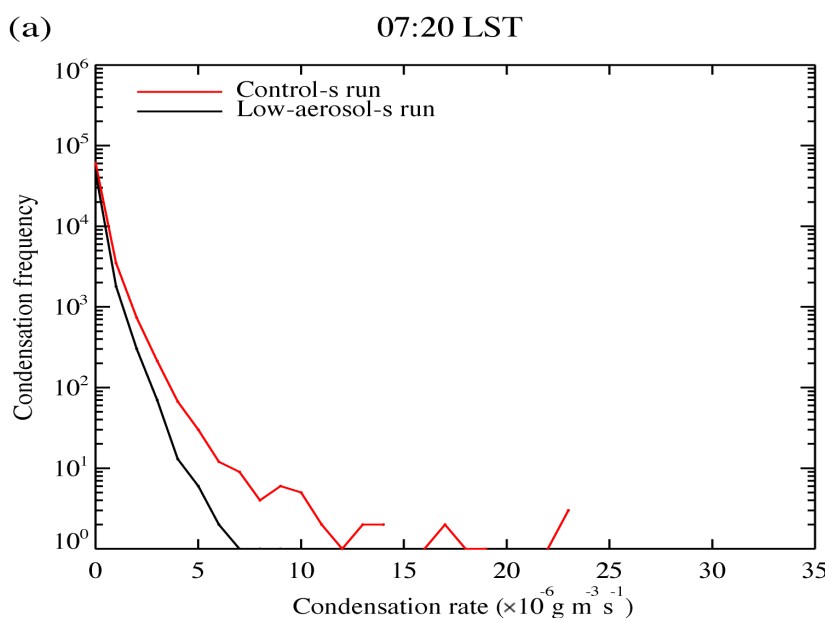

**(b)**  09:00 LST

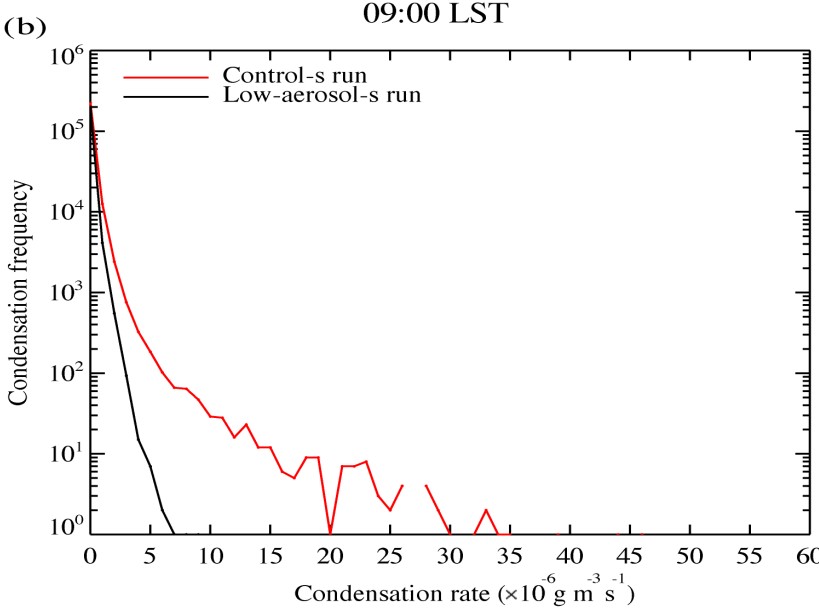


**Figure 10**





# Beijing case

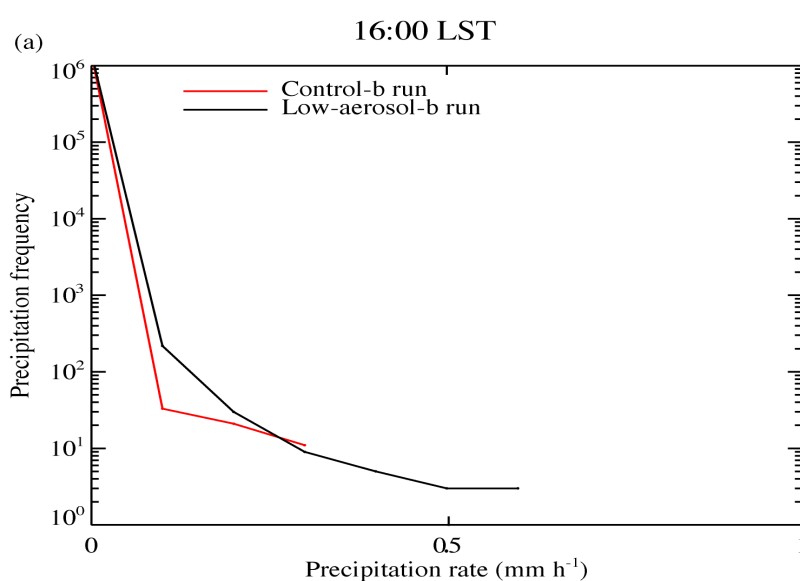

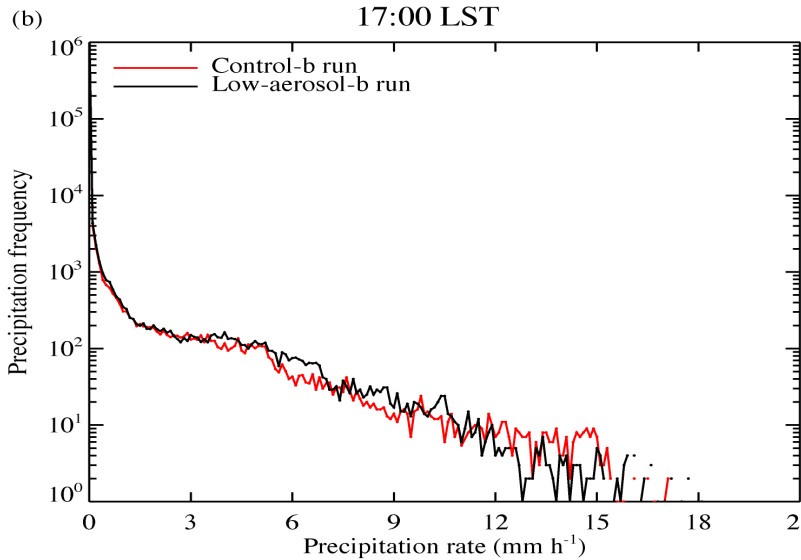

**Figure 11**





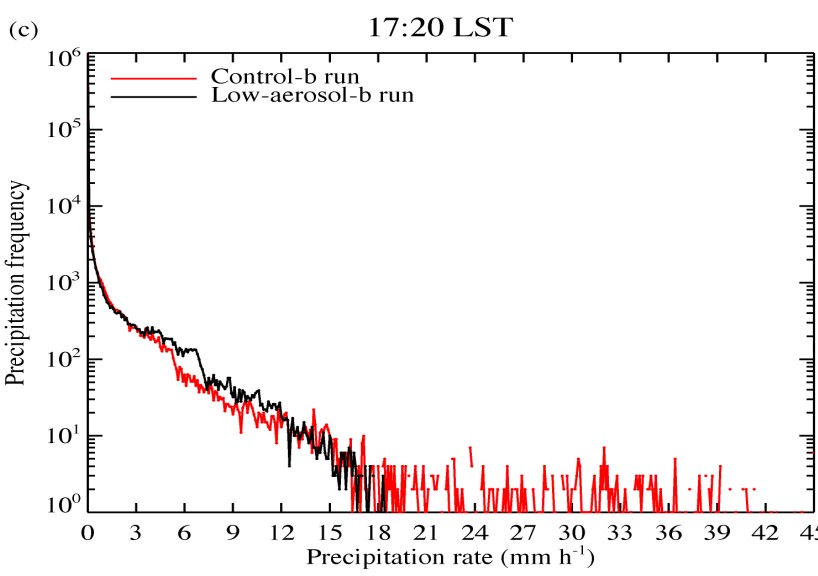

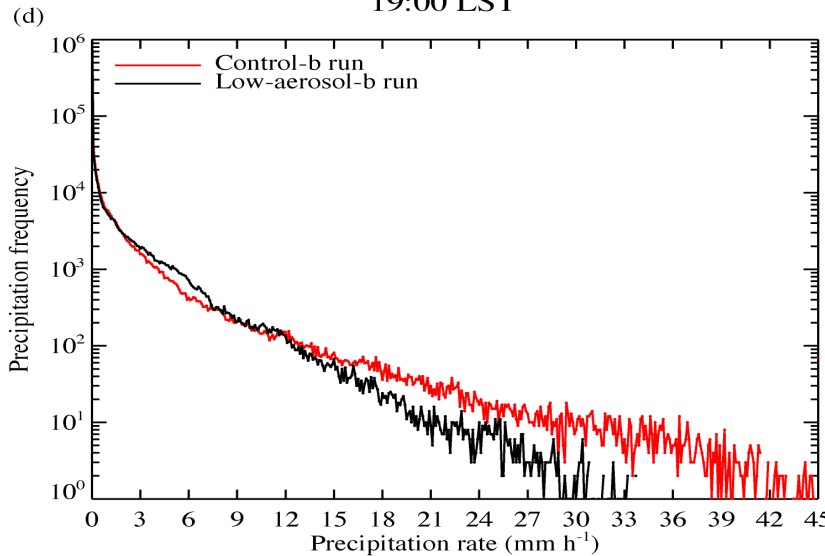

**Figure 11**





# Beijing case
## (control-b run minus low-aerosol-b run)

(a)                                    14:20 LST

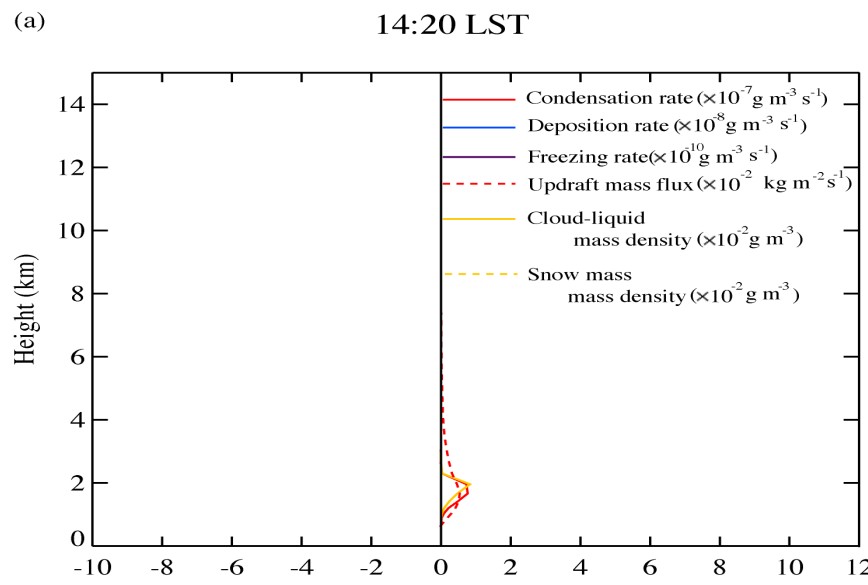

(b)                                    15:40 LST

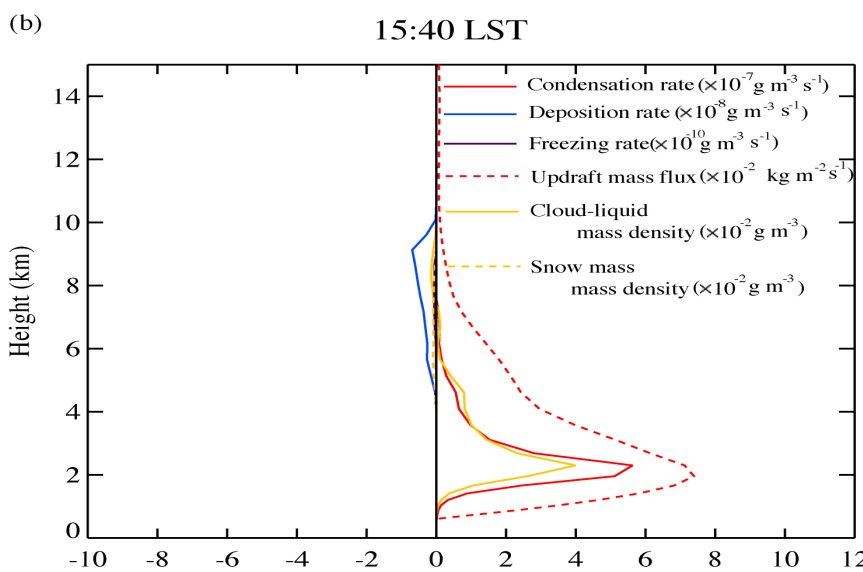

         **Figure 12**



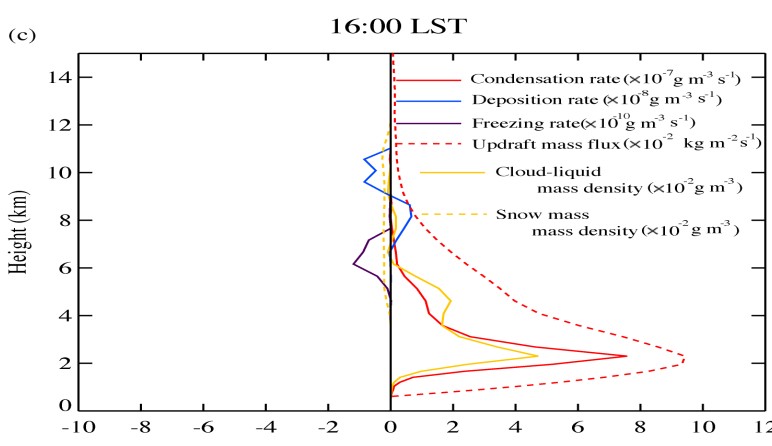

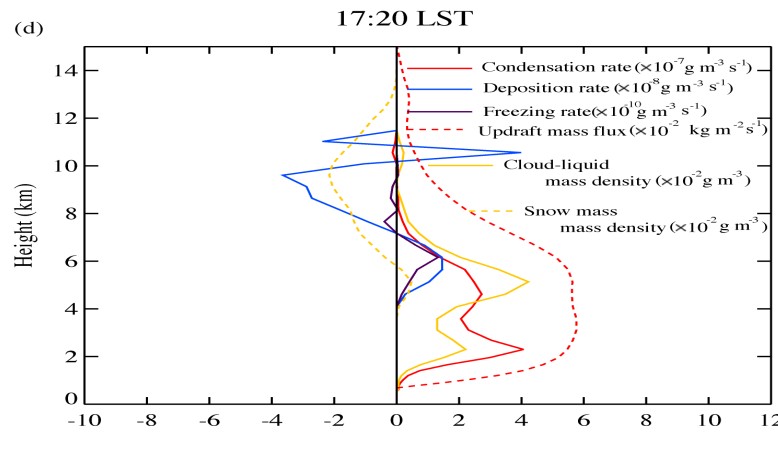

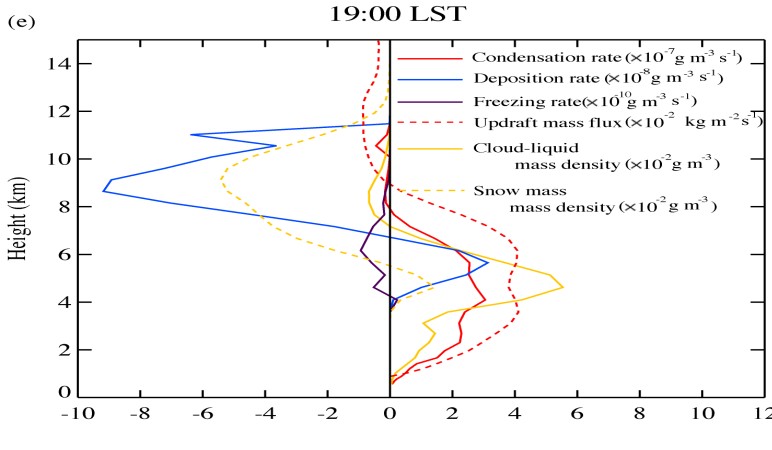


**Figure 12**



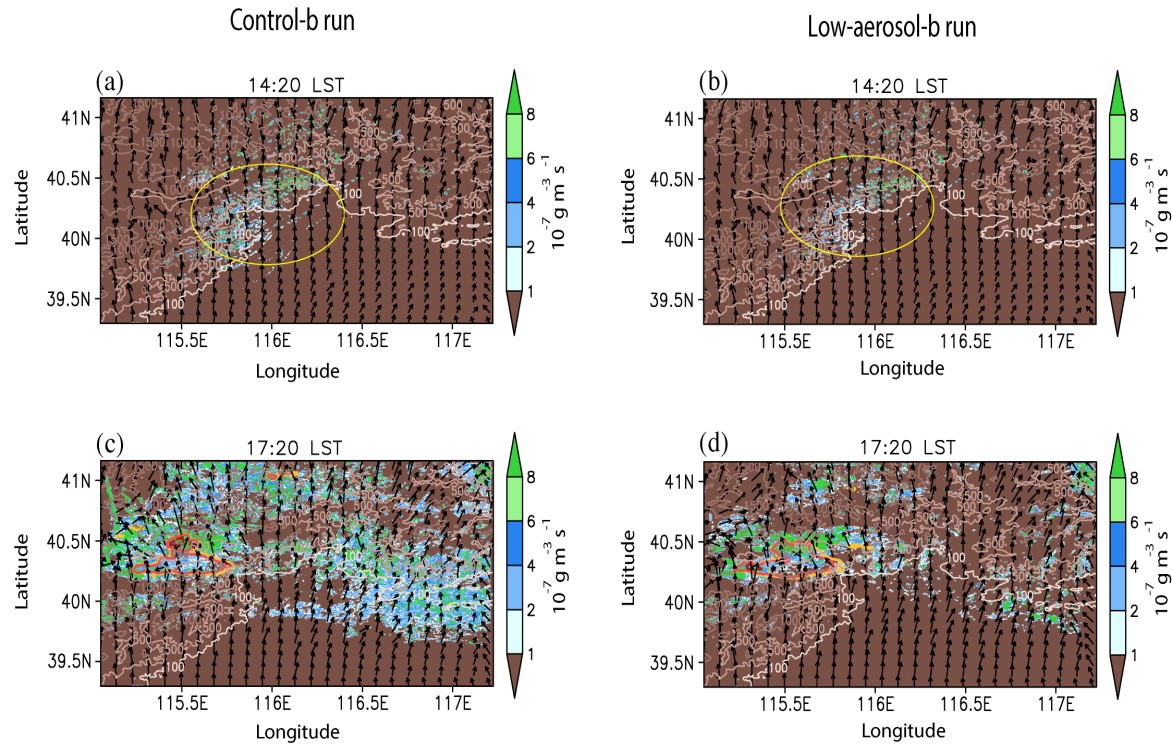


**Figure 13**





# Beijing case

(a)

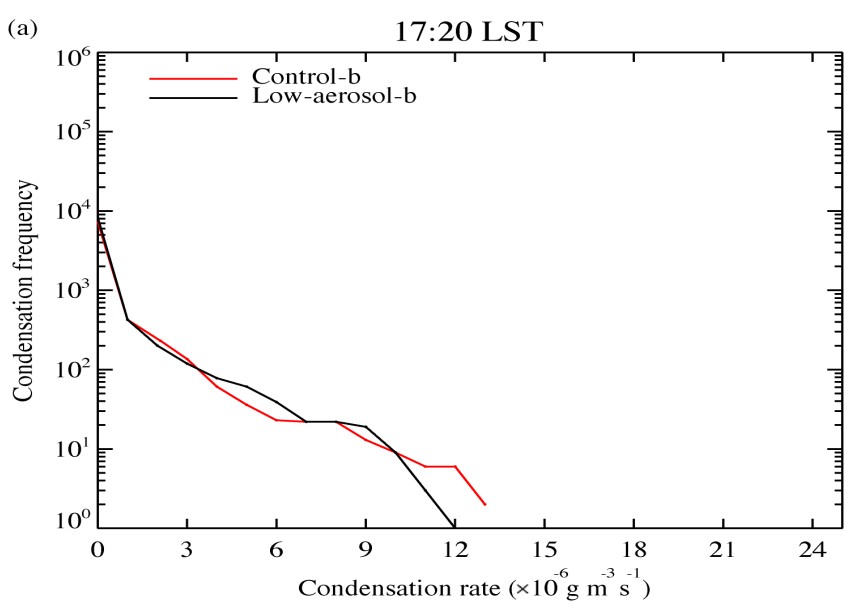

(b)

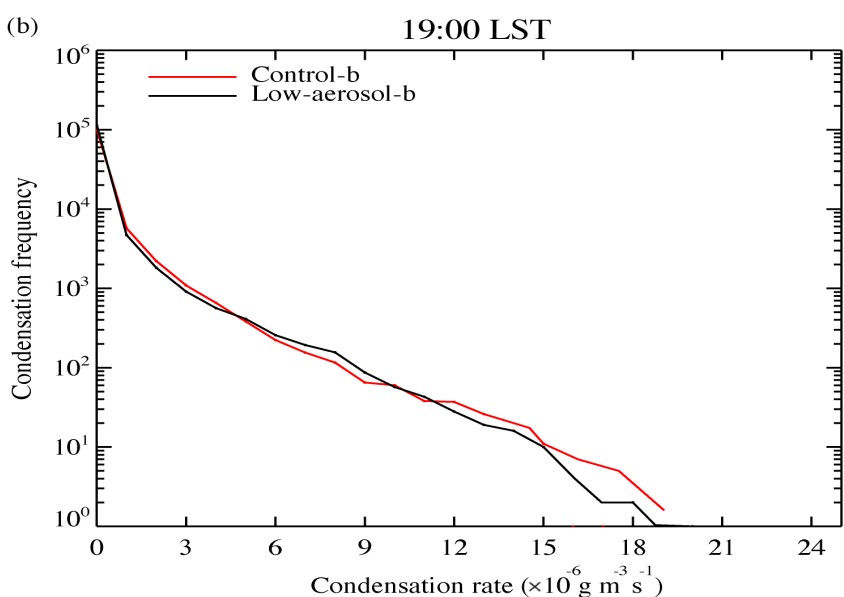


**Figure 14**








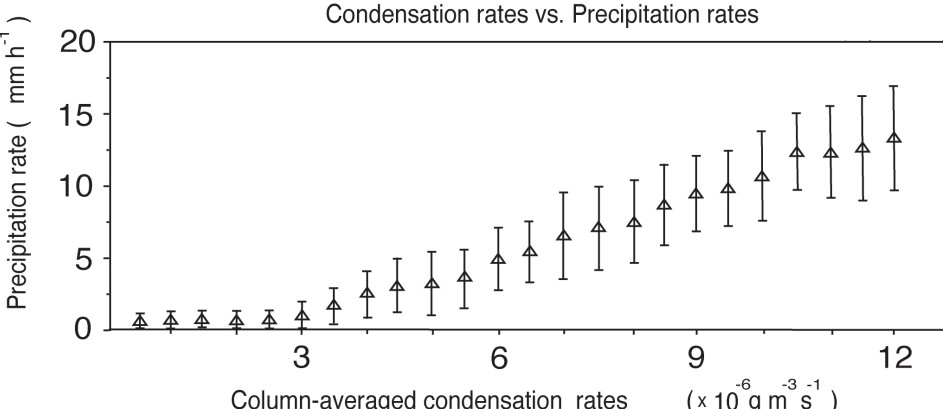


**Figure 15**

















**Figure 16**







**Figure 16**



# Beijing case

## 17:00 - 19:00 LST

(a)

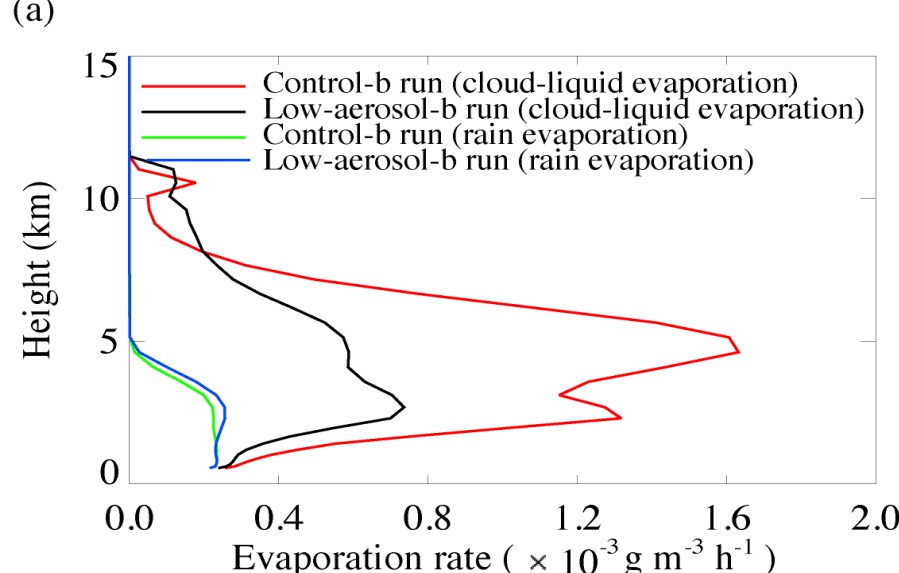

(b)

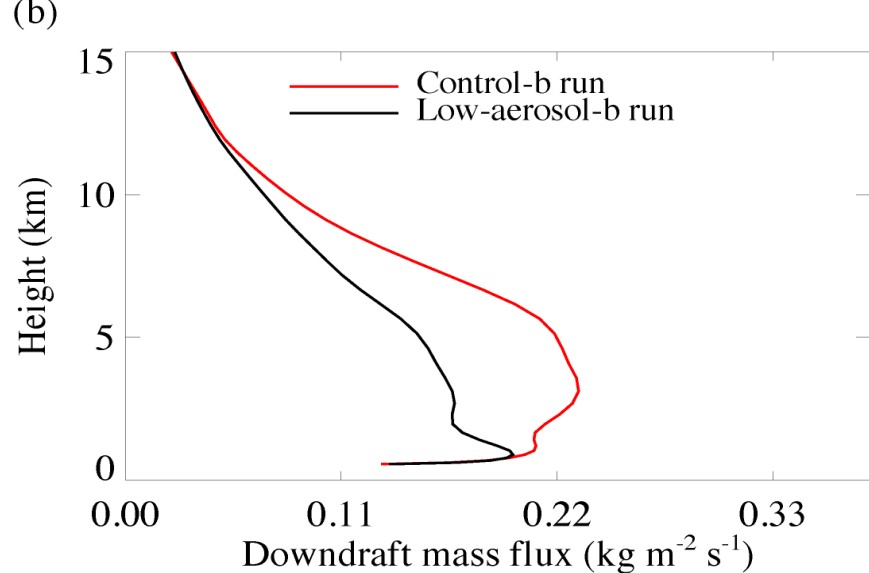

**Figure 17**





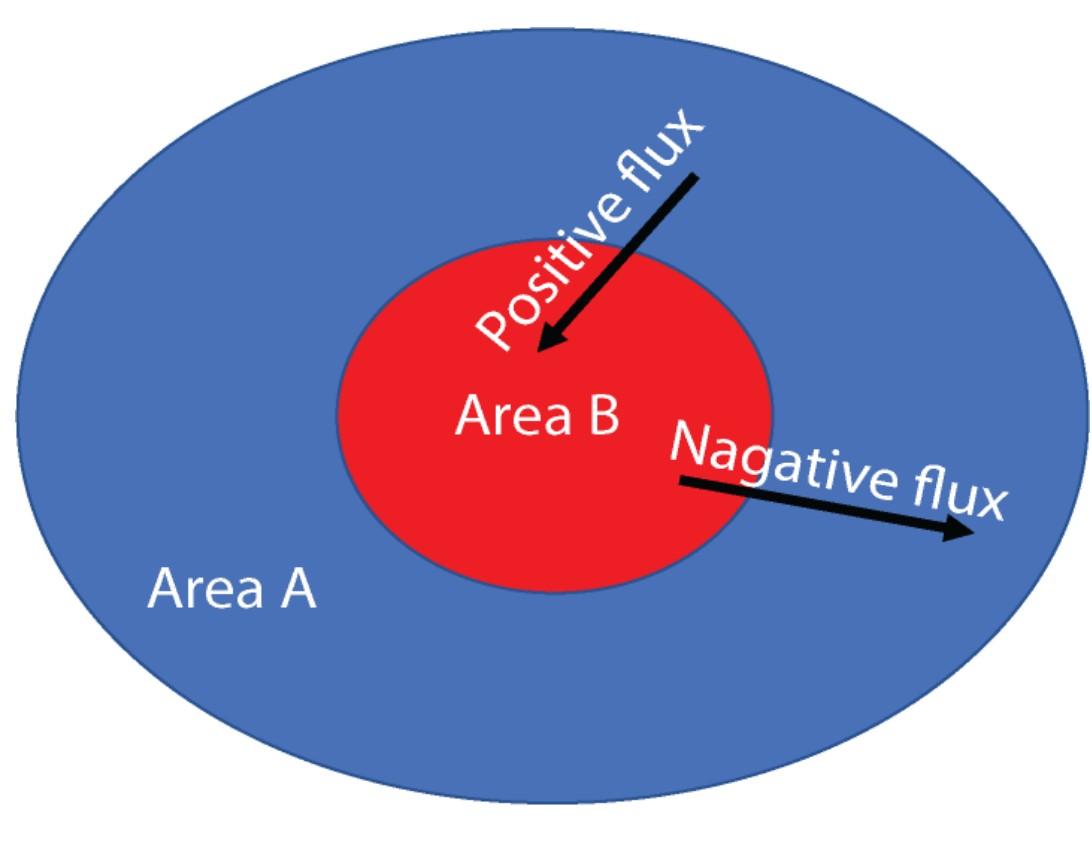


**Figure 18**



