# Peer review of "Examination of aerosol impacts on convective clouds and precipitation in two"

_Atmospheric Chemistry and Physics, 2021_

## Author Comment (AC1)

First of all, we appreciate the reviewer's comment and suggestion. In response to them, we have made relevant revisions to the manuscript. Listed below are our answers and the changes made to the manuscript according to the question and suggestion given by the reviewer. The comment of the reviewer (in black) is listed and followed by our responses (in blue).

This paper presents two cases studies of deep convective cloud systems. The authors perform simulations with a cloud-system resolving model to determine the impacts of increased concentrations of cloud condensation nuclei on these cloud systems, most specifically precipitation. The authors' analysis provides a valuable contribution to understanding aerosol-cloud interactions in deep convective cloud. However, some important details are missing from sections 2 and 3 of the manuscript. I discuss these in more detail below. I recommend the manuscript for publication, provided that my following concerns are addressed.

**General Comments:**

The authors do not currently provide the origin of any of the meteorological information presented in Sect. 2 or Fig. 1. I assume that this is sourced from a reanalysis. The authors must specify which reanalysis this information is sourced from and credit it appropriately (usually, with a citation). If some of the statements in this section are sourced from local observations, these should also be credited appropriately.

**The following is added:**

**(LL165-168 on p6)**

Note that synoptic features in Figures 1a and 1b are based on reanalysis data that are produced by the Met Office Unified Model (Brown et al., 2012) every 6 hours with a  $0.11^{\circ} \times 0.11^{\circ}$  resolution.

In multiple locations in the paper, the authors state that observations were interpolated and extrapolated to the model domain, without giving the method used. Given that there multiple valid methods of interpolating such data, as well a several methods that would be wholly inappropriate for this study, the authors should specify how the interpolation and extrapolation was done.

Observation data, such as PM and precipitation data, are extrapolated or interpolated into each time step and grid point. For these extrapolation and interpolation, the inverse distance weighting (IDW) method is used. The method is one of popular ones for the extrapolation and interpolation and detailed at

https://en.wikipedia.org/wiki/Inverse\_distance\_weighting. To indicate the IDW method used, the following is added:

**(LL222-224 on p8)**

In this study, the inverse distance weighting method is used for the extrapolation and interpolation of observation data including aerosol mass into grid points and time steps in the model.

For both case studies, differences in snow mass based on aerosol concentrations are discussed, but differences in hail and graupel are not mentioned. Is this because the differences are insignificant? Or was the mass of graupel and hail insignificant in all cases - in other words, all frozen water mass took the form of snow? Please specify this in the text.

In Figures 8 and 12, we added differences in the mass of other precipitable hydrometeors (i.e., raindrops and hail particles) between the runs. Accordingly, text is revised. Here, for the simplicity of the display, hail mass include graupel mass in Figures 8 and 12. Also, want to note that snow mass in Figures 8 and 12 includes the ice-crystal mass for the simplicity of the display.

The authors should consider whether it is necessary to show the full time evolution of the Soeul case, or whether they feel that some subplots of Figures 7, 8, and especially 9 can be moved to a supplement.

Figures 7b and 7c are removed and become supplementary Figures 1a and 1b. Text is revised accordingly.

Figures 9c, 9d, 9g, 9h, 9i, 9j, 9m, 9n, 9q, 9r, 9w,9x,9y, and 9z are removed and text is revised accordingly.

Regarding Figure 8, we believe that it is needed to keep all subplots, since we think that as seen in text, each of those subplots delivers important information on the evolution of differences in latent-heat processes and hydrometeors between the control-s and low-aerosol-s runs.

Technical comments:

In this study, the modifications in aerosol concentrations only affect clouds through their role as cloud condensation nuclei (CCN); direct effects of aerosol on radiative transfer are neglected and ice-nucleating particle (INP) concentrations are held constant. This is a fine experimental approach, and I don't wish to increase the scope of the paper. However, in both the abstract and conclusions, the authors never use the term CCN. I request that the authors change at least one use of "aerosol" to "cloud condensation nuclei" in both the abstract and the conclusions in order to make the focus of the paper more clear to a time-constrained reader.

**We revised the manuscript including abstract and conclusion to reflect points here. See text for details.**

Please also include the name of the model used in the abstract. This will help other researchers using the same model and researchers interested in comparing results between models to find your research.

The following is added:

**(LL57-59 on p3)**

These two areas are the Seoul and Beijing areas and the examination has been done by performing simulations using the Advanced Research Weather Research and Forecasting model as a cloud-system resolving model.

p2 line 29: has -> have

**Corrected.**

p2, lines 65-66: The first half of this sentence currently sounds like there is a decrease in cloud liquid which is not the case. Perhaps the authors should rephrase this as "...less cloud liquid forming raindrops..."?

**Corrected following the reviewer's suggestion here.**

p5, line 124: Where is the precipitation rate recorded? What is the source of this statement?

The maximum precipitation rate was observed at a location whose latitude is ~41N and longitude is ~116E. This location corresponds to the north-center part of the domain.

To clarify the observation source of precipitation rate, the following is added:

(LL159-162 on p6)

Here, similar to the situation in the Seoul area, precipitation in the Beijing area is measured by rain gauges in AWSs hourly with a spatial resolution that ranges from ~1 km to ~10 km. The Beijing area is marked by an inner rectangle in Figure 1b and Figure 2b and dots in the rectangle in Figure 2b mark the selected locations of rain gauges.

p6, line 164: Brown et al. (2012) does not appear in the reference list.

The following is added in the reference list:

Brown, A., Milton, S., Cullen, M., Golding, B., Mitchell, J., and Shelly, A.: Unified modeling and prediction of weather and climate: A 25-year journey, Bull. Am Meteorol. Soc. 93, 1865–1877, 2012.

p6, lines 178-180: Is there a reasoning behind this assumption? Specifically, is there a reason to assume that aerosol acting as CCN is larger over Beijing than Seoul?

In Seoul, only  $PM_{2.5}$  is available, while in Beijing, only  $PM_{10}$  is available. This is why in Seoul,  $PM_{2.5}$  is used to calculate aerosol number concentrations, while in Beijing  $PM_{10}$  is used to calculate those concentrations.

Most of CCN are in accumulation mode whose radius range is from ~0.1 to ~1 micrometer for both of the Seoul and Beijing cases as shown in Figure 3. Here, the radius range of the accumulation mode is determined by the AERONET data but not by PM data, hence, using  $PM_{10}$  in Beijing and  $PM_{2.5}$  in Seoul does not mean that CCN is larger over Beijing than over Seoul. Remember that for the AERONET-observed and - determined aerosol size distribution including accumulation mode,  $PM_{10}$  or  $PM_{2.5}$  is used to obtain aerosol number concentrations.

AERONET data show that at the surface, aerosols are composed of ammonium sulfate and organic compound. Fulvic acid, succinic acid and levoglucosan are the representative components of organic compound according to other studies such as Lance et al. (2005). These ammonium sulfate and organic compound are representative components of aerosols acting as CCN. Hence, PM2.5 and PM10, which are measured at the surface, are assumed to represent aerosols acting as CCN.

To clarify the reasoning behind the assumption pointed by the reviewer here, the corresponding text is revised as follows:

(LL242-245 on p9)

Since ammonium sulfate and organic compound are representative components of CCN, it is assumed that the mass of aerosols that act as CCN is represented by PM2.5 and PM10 for the Seoul and Beijing areas, respectively.

**Reference:**

Lance, S., A. Nenes, T. A. Rissman, Chemical and dynamical effects on cloud droplet number: Implications for estimates of the aerosol indirect effect, J. Geophys Res., 109, D22208, doi:10.1029/2004JD004596, 2005.

p7, lines 189-190: Why were these proportions chosen for the two sites?

The aerosol (chemical) composition is determined by data from the AERONET sites. The composition is based on the average AERONET data over the AERONET sites at a time point, which is 1 hour before the observed clouds start to form, for each of the Seoul and Beijing cases. Here, the AERONET data before clouds start to form are used, since it is found that the AERONET data are not available when clouds are present.

To clarify this, the corresponding text is revised as follows:

**(LL237-241 on p9)**

The AERONET data are averaged over the AERONET sites at 02:00 LST December 24th 2017 (13:00 LST July 27th 2015), which is 1 hour before the observed MCS forms, for the Seoul (Beijing) case. Based on the average data, it is assumed that aerosol particles are internally mixed with 70 (80) % ammonium sulfate and 30 (20) % organic compound for the Seoul (Beijing) case.

p7, line 197: absorbers -> absorber

**Corrected.**

p7, line 205: "with": do the authors mean within?

The corresponding text is revised as follows to make it clear based on this reviewer's comment and to reflect the other reviewer's comment:

**(LL261-264 on p9)**

The distribution parameters of the assumed shape of the size distribution of background aerosols in Figure 3a (3b) are those that are averaged over the AERONET

**sites at a time point, which is 1 hour before the observed MCS form, for the Seoul (Beijing) case.**

p7: Please give the details of the three aerosol modes for each case: number, median diameter, and geometric standard deviation. It might be most appropriate to give the number normalised by the PM2.5 or PM10 mass. How were they chosen? Are these fits to the AERONET data?

The parameters of the size distribution, which are median radius, geometric standard deviation for each mode and partition of aerosol number among modes, are obtained by fitting the AERONET data to the lognormal distribution.

As a way of giving the details of number, median radius, and geometric standard deviation as phrased by the reviewer here, the specific numbers of the parameters are given as follows:

**(LL256-261 on p9)**

Modal radius of the shape of distribution is 0.015 (0.012), 0.110 (0.085), and 1.413 (1.523)  $\mu$ m, while standard deviation of the shape of distribution is 1.28 (1.10), 1.54 (1.63), and 1.75 (1.73) for Aitken, accumulation and coarse modes, respectively, in the Seoul (Beijing) case. The partition of aerosol number, which is normalized by the total aerosol number of the size distribution, is 0.555 (0.612), 0.444 (0.387), and 0.001 (0.001) for Aitken, accumulation and coarse modes, respectively, in the Seoul (Beijing) case.

In the added text above, the aerosol number normalized by total aerosol number but not by PM mass is given, since the size distribution of interest here is about aerosol number but not aerosol mass.

p7, line 216: The aerosol decreases exponentially, but with what exponent?

The following is added:

**(LL271-273 on p10)**

With this exponential decrease, when the altitude reaches the tropopause, background concentrations of aerosols acting as CCN reduce by a factor of ~10 as compared to those at the PBL top.

p8, lines 212-213: Based on the previous text, I thought that the relative size distributions and aerosol compositions were held fixed for each case. Therefore, only the aerosol number concentrations should need to be interpolated or extrapolated,

right? Additionally, how was the extrapolation/interpolation done? Are concentrations linearly interpolated and extrapolated?

Aerosol composition and the shape of size distribution or size-distribution parameters (i.e., modal radius and standard deviation of each of Aitken, accumulation and coarse modes, and the partition of aerosol number among those modes) are held fixed in each of the Seoul and Beijing cases as the reviewer thought.

PM data are extrapolated or interpolated into each time step and grid point. For these extrapolation and interpolation, the inverse distance weighting (IDW) method is used. The method is one of popular ones for the extrapolation and interpolation and detailed at https://en.wikipedia.org/wiki/Inverse\_distance\_weighting. To indicate the IDW method used, the following is added:

**(LL222-224 on p8)**

In this study, the inverse distance weighting method is used for the extrapolation and interpolation of observation data including aerosol mass into grid points and time steps in the model.

Using the fixed aerosol composition and size-distribution parameters and extrapolated/interpolated PM data at each time step and grid point, the number concentration of background aerosols is determined at each time step and grid point. To clarify this, the corresponding text is revised as follows:

**(LL264-268 on p9-10)**

By using  $PM_{2.5}$  or  $PM_{10}$ , which is interpolated and extrapolated to grid points immediately above the surface and time steps, and based on the assumption of aerosol composition and size distribution above, the background number concentrations of aerosols acting as CCN are obtained for the simulation for each of the cases.

p8-9, lines 241-245: I find this sentence very confusing. The previous description by the authors seems to make it pretty clear that the background aerosol concentrations is a diagnostic field. For example, lines 218-220 "Once background aerosol properties (i.e., aerosol number concentrations, size distribution and composition) are put into each grid point and time step, those properties at each grid point and time step do not change during the course of the simulations." However, the phrasing of this sentence suggests that aerosol transport or advection is a process explicitly simulated by the model. Are the authors trying to say that, because the out-of-cloud aerosol concentrations are derived from observations, their spatial patterns and temporal

evolution will mimic advection that occurred in reality during the case study time period? Please clarify.

Yes, as the reviewer stated here, we wanted to say that because the out-of-cloud aerosol concentrations are derived from observations, their spatial patterns and temporal evolution will mimic advection that occurred in reality during the case study time period. Stated differently, we wanted to say that since aerosol concentrations, which are derived from observation, are at grid points and time steps of simulations, the spatiotemporal distributions of aerosol concentrations in those grid points and time steps in simulations can mimic those distributions and associated advection that occurred in reality during the simulation period.

The corresponding text is revised as follows:

```
(LL297-299 on p10-11)
```

This enables spatiotemporal distributions of background aerosols in the simulations to mimic those distributions that are observed and particularly associated with observed aerosol advection in reality.

p10, line 300: gird -> grid

**Done.**

p11, line 319: Do the authors have a reference for the AWS?

The following is added in the reference list:

King, J.: Automatic weather stations, available at https://web.archive.org/web/20090522121225/http://www.automaticweatherstation.com/index.html, 2009.

Associated with the reference, the following text is added:

**(LL143-145 on p5-6)**

Here, precipitation in the Seoul area is measured by rain gauges in automatic weather stations (AWSs) (King, 2009). The measurement is performed hourly with a spatial resolution that ranges from ~1 km to ~10 km.

(LL159-160 on p6)

Here, similar to the situation in the Seoul area, precipitation in the Beijing area is measured by rain gauges in AWSs hourly with a spatial resolution that ranges from  $\sim$ 1 km to  $\sim$ 10 km.

p12, lines 346-348 and Figure 6: why are the observations interpolated and extrapolated? How are they interpolated and extrapolated? Linearly? Why isn't the model subsampled to the times and locations of the observations?

For the extrapolation and interpolation of observation data, such as PM and precipitation data, the inverse distance weighting (IDW) method is used as stated in one of authors' responses above. The method is one of popular ones for the extrapolation and interpolation and detailed at <a href="https://en.wikipedia.org/wiki/Inverse\_distance\_weighting">https://en.wikipedia.org/wiki/Inverse\_distance\_weighting</a>

It is found that whether observation data are interpolated and extrapolated into model grid points and time steps or model outputs are interpolated and extrapolated into observation sites and time points does not affect the qualitative conclusions drawn from Figure 6.

p12-13, lines 365-367: The authors should be more precise here. For what range is the difference between control-s and low-aerosol-s greater than a factor of 10? To my eye, this does not seem to occur below 11 mm h-1.

At 11.4 mm hr-1, the cumulative precipitation frequency is 10 in the control-s run and the frequency is 1 in the low-aerosol-s run. To reflect this, the corresponding text is revised as follows:

**(LL422-424 on p15)**

In particular, for the precipitation rate of 11.4 mm h-1, there is an increase in the cumulative frequency by a factor of as much as ~10 in the control-s run.

p17, lines 497-498: The authors should be more precise here. They should use the greatest whole number for which the statement is true, instead of 12 mm hr-1. From looking at Fig. 6b, the two precipitation frequencies don't seem to differ by a factor of 10 for precipitation rates less than 27 mm hr-1.

It is found that for the precipitation rates of 28.1 and 30.0 mm hr-1 each, the cumulative precipitation frequency increases by a factor of ~10 in the control-b run. The corresponding text is revised as follows:

(LL584-585 on p20)

**Particularly, for the precipitation rates of 28.1 and 30.0 mm hr-1, the cumulative frequency increases by a factor of as much as ~10.**

p17, lines 517-520: This sentence is confusing. It sounds like the authors are saying that the distinctive pattern (control-b greater for precipitation rates <2 mm hr-1 or >22 mm hr-1, low-aerosol-b greater for precipitation rates between 2 and 12 mm hr-1) is emerging at this time. However, they already stated that the pattern started to emerge at 17:00. Are they authors simply saying that the differences between the two simulations have become more pronounced? Are they trying to state that control-b becomes greater for precipitation rates <2 mm hr-1 at this time, while the relationship between control-b and low-aerosol-b is unchanged for greater precipitation rates? Or are they trying to state that the cumulative frequency distribution of control-b has changed from 17:00 to 17:20, while the cumulative frequency distribution of low-aerosol-b remained relatively unchanged during this time period?

**Here, we are trying to say that as the reviewer stated, the control-b run becomes greater for precipitation rates <2 mm hr-1 at this time, while the relationship between the control-b and low-aerosol-b runs is unchanged for greater precipitation rates.**

At the last time step, as seen in Figure 6b, the frequency of comparatively heavy precipitation whose rates are higher than ~12 mm hr-1 rises significantly in the controlb run as compared to that in the low-aerosol-b run. Below ~2 mm hr-1, there is also the greater precipitation frequency in the control-b run than in the low-aerosol-b run. Unlike the situation for precipitation rates above ~12 mm hr-1 and below ~2 mm hr-1, for precipitation rates from ~2 to ~12 mm hr-1, the control-aerosol-b run has the lower precipitation frequency than in the low-aerosol-b run. In the corresponding text, we want to inform readers of time points when qualitative patterns of these differences between the runs, which are as shown in Figure 6b, emerge. For this, first, we want to say that at 17:00 LST, the pattern of the higher (lower) cumulative precipitation frequency in the control-b run, as compared to that in the low-aerosol-b run, for precipitation rates higher than ~12 mm hr-1 (between ~2 and ~12 mm hr-1) starts to occur. Then, we want to say that at 17:20 LST, the pattern of the higher cumulative precipitation frequency in the control-b run, as compared to that in the low-aerosol-b run, for precipitation rates lower than ~2 mm hr-1 starts to occur, while the pattern of the higher (lower) cumulative precipitation frequency in the control-b run, which is established at 17:00 LST, for precipitation rates higher than ~12 mm hr-1 (between ~2 and ~12 mm hr-1) maintains as time progresses from 17:00 LST to 17:20 LST. Here, we are not interested in the temporal evolution of absolute values of the frequency in the runs but in when the pattern of higher or lower frequency in the control-b run or in the low-aerosol-b run at specific ranges of the precipitation rates starts to occur.

Based on the argument here, text pointed out here is revised as follows:

**(LL601-610 on p20-21)**

As time progresses to 17:00 LST, the maximum precipitation rate increases to ~17 mm hr-1 and the higher (lower) cumulative precipitation frequency over precipitation rates higher than ~12 mm hr-1 (between ~2 and ~12 mm hr-1) in the control-b run than in the low-aerosol-b run, which is described above as shown in Figure 6b for the last time step, starts to emerge (Figure 11b). At 17:20 LST, the higher frequency for precipitation rates below 2 mm hr-1 in the control-b run, which is also described above as shown in Figure 6b for the last time step, starts to show up, while the higher (lower) frequency for precipitation rates below 1 mm hr-1 in the control-b run, which is also described above as shown in Figure 6b for the last time step, starts to show up, while the higher (lower) frequency for precipitation rates higher than ~12 mm hr-1 (between ~2 and ~12 mm hr-1) in the control-b run, which is established at 17:00 LST, maintains as time progresses from 17:00 LST to 17:20 LST (Figure 11c).

p18, lines 541-542: This sentence does not make sense as currently written. I think that this sentence can be simplified to "This leads to more condensation in the control-b run."

Simplified following the suggestion here.

p18, lines 550-551: Why do the authors specify that the differences are at altitudes "with non-zero differences in deposition rates between the runs"? This is not only redundant, it makes the sentence confusing.

"with non-zero differences in deposition rates between the runs" is removed. The corresponding text is revised as follows:

**(LL654-657 on p22)**

Due to stronger updrafts, which are mainly ascribed to more condensation, deposition rates start to be higher at altitudes between ~7 and ~9 km and freezing rates are higher at altitudes between ~4 and ~6 km in the control-b run with the time progress from 15:40 LST to 16:00 LST (Figure 12c).

Also, other text with "with non-zero differences in deposition rates between the runs" is removed and revised.

p19, lines 553-554: As above, why specify that the differences are where the differences are non-zero?

"with non-zero differences in deposition rates between the runs" is removed throughout the manuscript. The corresponding text is revised as follows:

**(LL660-663 on p22)**

At 17:20 LST, overall, freezing rates are lower at altitudes between ~4 and ~8 km, while overall, snow and hail mass is still lower, and droplet mass is still higher in the control-b run (Figure 12d).

p23, lines 675-678: see note regarding p24, lines 727-734 below.

Text pointed out here is removed and the following is added in the original discussion for Figure 15:

**(LL735-739 on p25)**

It is found that this correspondence between condensation and precipitation rates is valid whether analyses to construct Figure 15 are repeated only for a time point at 16:30 LST or for a period between 16:30 and 17:00 LST. These time point and period are related to analyses of the moist static energy as described in Section e below.

p24, lines 716-719: Why divide by the total number of grid cells? If averaging is to be done, it seems more intuitive to average only over cells containing the boundary between areas A and B. An analogous variable would be the cloud droplet number concentration: when an average is taken, typically only cloudy grid cells would be included in the average. I recommend using the total (net) flux instead. The text would be simpler if you discussed the total flux instead, and it would not alter your conclusions.

**Following the comment here, the corresponding text is revised by replacing the domain-averaged fluxes with total fluxes in text.**

p24, lines 727-734: This is repetitive with respect to lines 675-678, and with respect to the original discussion of Fig. 15 on pages 20-21. It would be better to instead note during the original discussion of Fig. 15 that the calculation was repeated for the restricted time periods, and the correspondence between the specified condensation rates and precipitation rates were found to be valid for the restricted time periods. Then it would not be necessary to repeat so much text multiple times.

**Text pointed out here is removed and the following is added in the original discussion for Figure 15:**

**(LL735-739 on p25)**

It is found that this correspondence between condensation and precipitation rates is valid whether analyses to construct Figure 15 are repeated only for a time point at 16:30 LST or for a period between 16:30 and 17:00 LST. These time point and period are related to analyses of the moist static energy as described in Section e below.

p28, lines 833-836: The authors should either change "aerosol-induced" to "CCNinduced" for this sentence, or add the qualification that this is at fixed INP concentrations. The results may have been different if INP concentrations were reduced by the same factor as CCN concentrations, and this effect would still be an aerosol-induced variation in freezing.

**The corresponding sentence is revised as follows:**

**(LL931-934 on p31)**

Here, we find that contrary to the traditional understanding, the role of variation of freezing, which is induced by the varying concentration of aerosols acting as CCN but not INPs, in precipitation is negligible as compared to that of condensation and deposition in both of the cases.

p29, line 863: please remove "the" between "steal" and "more".

**Done.**

Throughout the discussion and conclusions, the authors refer to "strong clouds". Do the authors mean vertically-thick clouds, or high-water-content clouds, or are they using some other metric for strength?

As described in Section e, titled "moist static energy", clouds in area B have greater condensational heating and associated stronger updrafts than those in aera A. Based on these stronger updrafts, we say clouds in area B are stronger than those in area A. Stated differently, clouds in area A are strong and those in area B are less strong. Due to more condensation, condensational heating and associated stronger updrafts, "strong clouds" in area B are vertically thicker with higher water content (or cloud mass) than "less strong clouds" in area A or "less strong clouds" in area A are vertically less thick with lower water content than "strong clouds" in Area B.

In the discussion and conclusions, strong clouds mean clouds in area B and less strong clouds mean clouds in area A. Due to less condensation, condensational heating and

associated weaker updrafts in these less strong clouds in area A, these "less strong clouds" are vertically less thick with lower water content than "strong clouds" in area B.

Based on the argument in the above two paragraphs, to classify clouds to "strong clouds" and "less strong clouds", we use the metric of all of updrafts, cloud thickness and water content.

To clarify points here, the following is added:

(LL899-901 on p30)

note that these strong clouds here involve stronger updrafts via greater condensational heating as described in Section e above and this enables these clouds to be thicker and have higher cloud mass than these less strong clouds.

Figure 1: Is the potential temperature shown at the 850 hPa height, like the wind, or at a different vertical level?

Yes, the potential temperature as well as wind and geopotential height is at the 850 hpa height. Accordingly, the corresponding part of the caption is revised as follows:

Wind (m s-1), equivalent potential temperature (K), and geopotential height (m) at 850 hPa level over Northeast Asia

Fig. 9 and 13: The wind vectors are not mentioned in the figure captions. Are these at the surface?

Yes, they are at the surface. This is indicated in the corresponding captions. Also, scaled wind vector is added on the top right corner of each panel in Figures 9 and 13.

Fig. 15: It should be specified in the caption that data from the beginning to the simulation to 17:20 was used for this figure.

The caption is revised as follows:

Figure 15. Mean precipitation rates corresponding to each column-averaged condensation rate for the period between 14:00 and 17:20 LST in the control-b run. One standard deviation of precipitation rates is represented by a vertical bar at each condensation rate.

---

## Author Comment (AC2)

First of all, we appreciate the reviewer's comment and suggestion. In response to them, we have made relevant revisions to the manuscript. Listed below are our answers and the changes made to the manuscript according to the question and suggestion given by the reviewer. The comment of the reviewer (in black) is listed and followed by our responses (in blue).

Effect of CCN concentration on convective precipitation formation in two case studies is presented. State of the art cloud resolving model with detailed representation for cloud microphysics is employed. The analysis highlights the role of aerosol in affecting the formation of heavy precipitation on both studied locations. In the Beijing case the analysis related to changes in surface precipitation is interesting. Especially how precipitation pattern changes towards stronger precipitation. I guess the computational cost of the simulations is relatively high, and thus ensembles with more variability in the aerosol properties cannot be employed to make the analysis stronger. Based on one pair of simulations, the uncertainty related to the overall strength of processes is high as small change in initial conditions could affect the relative aerosol effect a lot. However, how different processes and feedbacks work needs to be analyzed in more detail before I can recommend the manuscript to be accepted to be published in ACP.

It seems like the changes in condensation rate are now presented as the only reason for the increased updrafts. It goes also in opposite as stronger updrafts automatically increases also condensation rate. Increase in the aerosol concentration followed by increase in droplet concentration should not produce very significant change in the condensation rate as droplet concentration are expected to be relatively high even in the low aerosol simulations. Presenting the droplet number concentration would be thus highly beneficial. I would expect it is more likely that delayed formation of precipitation in warm processes is affecting more on liquid water content than the increased condensation through enhanced cloud droplet formation. This could also affect cloud radiative properties. Also, there is no discussion on how much column averaged condensation rates are affected by the change in the fraction of cloudy grid boxes or how for example the cloud top height is changing due to aerosol changes and cloud invigoration. The increase in condensation rate can be directly connected to increase in the precipitation like is done in the manuscript, but connecting the change in condensation rate to increased droplet number concentration requires more evidence. For example, what would happen if you just increase the condensation coefficient in low aerosol case?

We believe that this general comment is closely linked to the following specific comments below:

So, we suggest that the reviewer should look at our responses to these three comments as a way of finding our response to this general comment.

In the abstract the statement "In both of the areas, aerosol-induced changes in freezing play a negligible role in aerosol-precipitation interactions as compared to the role played by aerosol-induced changes in condensation and deposition." is quite strong as the number of ice particles is not affected by aerosol. Thus, through the manuscript, I suggest changing the wording to be about the change in CCN instead the change in aerosol.

We changed wording following the suggestion here

The analysis of gust fronts is somewhat separated from the rest of the analysis. What is the role of droplet evaporation through the simulations in driving the instability? Especial when compared against the enhanced condensation from increased droplet concentration?

Aerosol-induced enhancement of the frequency of precipitation with rates above 12 mm hr$^{-1}$ starts to emerge at 17:00 LST as described in Section a for the Beijing case via mechanisms associated with exchanges of the moist static energy between areas A and B as described in Section e "moist static energy". Not only the initial small differences in the frequency of precipitation with rates above 12 mm hr$^{-1}$ between the control-b and low-aerosol-b runs at 17:00 LST enhance substantially but also the maximum precipitation rate and its differences between the runs enhance substantially particularly during the period between 17:00 and 19:00 LST as seen in Figures 11b, 11c and 11d. Note that the establishment of the differences in the precipitation (cumulative) frequency and the maximum precipitation at 19:00 LST between the runs, followed by this substantial enhancement during the period between 17:00 and 19:00 LST, is basically equivalent to the establishment of those differences at the last time step of the runs as described in Section a for the Beijing case. Hence, we are interested in mechanisms controlling this substantial enhancement during the period between 17:00 and 19:00 LST and this is why we analyzed gust fronts and associated processes such as evaporation during the period between 17:00 and 19:00 LST as shown in Section d "evaporation and gust fronts" and why we do not take interest in the role of evaporation before 17:00 LST and after 19:00 LST.

Aerosol-induced increases in droplet evaporation make positive feedbacks with aerosol-induced increases in condensation. As described in Section b for the Beijing case, at the initial stage of cloud development before 17:00 LST, due to aerosol-induced increases in the nucleation of droplets and associated increases in the integrated surface of droplets, there are aerosol-induced increases in condensation. Then, as shown in Section d "evaporation and gust fronts", these aerosol-induced increases in condensation in turn enhance the amount of cloud liquid as a source of evaporation and then the droplet evaporation. The enhanced droplet evaporation in turn intensifies the gust fronts more, and this more intensified gust fronts in turn intensify updrafts and enhance condensation more in the control-b run than in the low-aerosol-b run during the period between 17:00 and 19:00 LST.  In particular, the more enhanced condensation due to the more intensified gust fronts during the period between 17:00 and 19:00 LST in the control-b run contributes to the substantially increasing differences in the frequency of precipitation with rates above ~ 12 mm hr$^{-1}$ between the runs, the substantially increasing maximum precipitation and its differences between the runs during the period between 17:00 and 19:00 LST.

To describe overall roles of droplet evaporation through the simulations or the whole simulation period, let me come up with the classic theory of evaporation and associated instability as follows. The following is basically about well-known classic theory, hence, we don't discuss it in text:

As well known and well described in Houze (1993) and Weisman and Klemp (1982), in the presence of wind shear, the evaporation of condensed liquid or droplets detrained to unsaturated areas and associated evaporative cooling do not align with condensation and associated condensational heating in the vertical direction. When condensational heating and evaporative cooling are vertically aligned at a similar horizontal location, the cooling stabilizes air and weakens convection. However, in the presence of wind shear as in the Beijing case, areas with evaporative cooling do not align with those with condensation heating in the vertical direction, and instead, those areas with evaporative cooling are located in different places in the horizonal domain as compared to those with condensational heating. This enables those areas with evaporative cooling can form gust fronts. These gust fronts lift surrounding warm air and generate subsequent convection via the forced-convection mechanism, and this in turn generate subsequent updrafts, condensation and precipitation.  More condensation and cloud liquid as a source of evaporation induce more evaporative cooling and more intensified gust fronts in the control-b run than in the low-aerosol-b run. This formation of gust fronts and their CCN-induced intensification exist throughout the simulation period. However, roles of this formation and intensification of gust fronts in precipitation with rates above 12 mm hr$^{-1}$ and the maximum precipitation are not significant before 17:00 LST when the MCS is at its initial stage and

after 19:00 LST when the MCS enters its decaying stage, and become significant to induce above-described substantial enhancement of differences in the frequency of precipitation with rates above 12 mm hr$^{-1}$ and the maximum precipitation rate between the runs during the time period between 17:00 and 19:00 when the MCS is at its mature stage.  During the mature stage, wind shear is stronger, hence, areas with evaporative cooling are located in different places in the horizonal domain as compared to those with condensational heating in a more effective way than during the initial and decaying stages. Therefore, during the mature stage, the formation of gust fronts is more efficient or gust fronts are formed more fully, and this enables CCN-induced more evaporative cooling to affect gust fronts more effectively. Hence, roles of the formation and intensification of gust fronts in precipitation are more substantial during the mature stage than during the initial and decaying stages.

To make a better connection between Section d about gust fronts and the other sections, a part of text in Section d is revised as follows:

(LL765-776 on p26)

As described above, the more droplet nucleation and greater integrated droplet surface induce more condensation before 17:00 LST in the control-b run. This and lower efficiency of collision and collection among droplets enable the control-b run to have a larger amount of cloud liquid or droplets as a source of evaporation. This in turn enables more droplet evaporation, more associated cooling and stronger downdrafts, although less rain evaporation is in the control-b run particularly for the period from 17:00 LST to 19:00 LST (Figure 17). More evaporation of droplets and associated stronger downdrafts with higher concentrations of aerosols acting as CCN have been shown by the numerous previous studies (e.g., Tao et al., 2007; Tao et al., 2012; Khain et al., 2008; Lee et al., 2018).
     During the period between 17:00 and 19:00 LST, with the development of convergence or the gust front, as mentioned above, the maximum precipitation rate increases from ~ 17 (17) to ~ 45 (33) mm hr$^{-1}$ in the control-b (low-aerosol-b) run (Figure 11).

(LL780-787 on p26)

Over the period from 17:00 LST to 19:00 LST, stronger downdrafts and associated stronger outflow generate a stronger gust front and more subsequent condensation in the control-b run. This enhances the small initial difference, which is at 17:00 LST, in the frequency of precipitation with rates above ~12 mm hr-1 between the runs substantially as time progresses from 17:00 LST to 19:00 LST (Figure 11). Associated with this, with the time progress, the nearly identical maximum precipitation rate

between the runs at 17:00 LST turns into the significantly higher maximum precipitation rate in the control-b run than in the low-aerosol-b run (Figure 11).

Specific comments:

Line 59: "With increasing aerosol loading or concentrations, cloud-particle sizes and autoconversion, which represent cloud microphysical properties, can be changed." Autoconversion is not a microphysical property but a parameterized process representation to create precipitation from cloud droplets. As you have binned microphysics employed, I would expect there is no need for the autoconversion.

The word "autoconversion" is removed in text and text is revised accordingly. However, just want to mention that in the microphysical scheme adopted in this study, for the calculation of cloud liquid content and rain water content which are variables in the WRF model, drops whose radius is smaller (greater) than 40 micrometer are treated to be droplets (raindrops), although there are no separate size distributions between droplets and raindrops. Stated differently, there is only one size distribution for drops and in this size distribution, drops whose radius is smaller (greater) than 40 micrometer are treated to be droplets (raindrops) for the calculation of the WRF variables that are used in other schemes such as radiation and PBL schemes. With this treatment, autoconversion is defined to be a process where droplets, which are drops with radius smaller than 40 micrometers, collide and collect each other to form raindrops, which are drops with radius greater than 40 micrometers. This definition of autoconversion is that of autoconversion in authors' responses below, although the word "autoconversion" is not in the manuscript anymore.

Line 192: "based on the fact that aerosol composition does not vary significantly over the domain and during the whole period with the observed clouds." How do you know this? AERONET does not provide data in cloudy conditions.

We re-checked the AERONET data used and found that we tried to collect the AERONET data from a time point 1 hour before the observed clouds form to the end of the simulation period. However, due to the reason stated by the reviewer here, we found that the AERONET data from a time point when the clouds form to the end of the simulation period were not taken into account for calculating aerosol composition and size distribution and only those data, which are at a time point 1 hour before the observed clouds form, are considered to obtain aerosol composition and size distribution.

Accordingly, we corrected text as follows:

(LL237-241 on p9)

The AERONET data are averaged over the AERONET sites at 02:00 LST December 24th 2017 (13:00 LST July 27th 2015), which is 1 hour before the observed MCS forms, for the Seoul (Beijing) case. Based on the average data, it is assumed that aerosol particles are internally mixed with 70 (80) % ammonium sulfate and 30 (20) % organic compound for the Seoul (Beijing) case.

(LL261-264 on p9)

The distribution parameters of the assumed shape of the size distribution of background aerosols in Figure 3a (3b) are those that are averaged over the AERONET sites at a time point, which is 1 hour before the observed MCS form, for the Seoul (Beijing) case.

Line 202: Aitken mode, not nucleation

Corrected.

Line 223: IN concentration seems high, what is the temperature dependence for heterogeneous nucleation and what is the actual ice number concentration in the simulations?

To describe how nucleation processes are represented, the following is added:

(LL190-197 on p7)

A cloud-droplet nucleation parameterization based on Köhler theory represents cloud-droplet nucleation. Arbitrary aerosol mixing states and aerosol size distributions can be fed to this parameterization. To represent heterogeneous ice-crystal nucleation, parameterizations by Lohmann and Diehl (2006) and Möhler et al. (2006) are used. In these parameterizations, contact, immersion, condensation-freezing, and deposition nucleation paths are all considered by taking into account the size distribution of INPs, temperature and supersaturation. Homogeneous droplet freezing is considered following the theory developed by Koop et al. (2000).

As described in the added text, the heterogeneous nucleation is represented by parameterizations by Lohmann and Diehl (2006) and Möhler et al. (2006). The details of these parameterizations are as follows and these details are not presented in text for the sake of brevity of text:

In Lohmann and Diehl's (2006) parameterizations, the contact activation is parameterized as follows:

$$\frac{dN_{CNT}}{dt}(m^{-3}s^{-1}) = m_{io}D_{ap}4\pi r_m N_{a,cnt}\frac{N_l^2}{\rho_a q_c} \qquad (1)$$

where $\frac{dN_{CNT}}{dt}$ is the rate of ice-crystal number production via contact freezing, $m_{io}$ is the original mass of a newly formed ice crystal, $\rho_a$ air density, $q_c$ the mass mixing ratio of droplets, $N_l$ the number mixing ratio of droplets, $D_{ap}$ (m² s⁻¹) is the Brownian aerosol diffusivity, $r_m$ is volume mean droplet radius and $N_{a,cnt}$ (m⁻³) is the number concentration of contact nuclei. In Lohmann and Diehl's (2006) parameterizations, immersion and condensation-freezing activation is parameterized as follows:

$$\frac{dN_{IMM}}{dt}(m^{-3}s^{-1}) = N_{a,imm}\exp(T_0 - T)\frac{dT}{dt}\frac{\rho_a q_c}{\rho_w} \qquad (2)$$

where $\frac{dN_{IMM}}{dt}$ is the rate of ice-crystal number production via immersion and condensation freezing, $\rho_w$ water density, $T$ air temperature, $T_0$ freezing air temperature. $N_{a,imm}$ (m⁻³) the number concentration of immersion and condensation nuclei. For deposition nucleation, Möhler et al.'s [2006] parameterization calculates the deposition nucleation as follows:

$$\frac{dN_{DEP}}{dt}(m^{-3}s^{-1}) = N_{a,dep}(\exp[a(S_i - S_0)] - 1) \qquad (3)$$

where $\frac{dN_{DEP}}{dt}$ is the rate of ice-crystal number production via depositional freezing, $S_i$ saturation ratio with respect to ice, $a$ and $S_0$ are non-dimensional empirical constants determined by chamber experiments, which are dependent on aerosol properties. Here, $a$ and $S_0$ are set to 4.77 and 1.07, respectively, based on experiments for dust.

$N_{a,dep}$ is the number concentration of deposition nuclei (m⁻³). This parameterization is applied to grid points with no cloud liquid to make sure only deposition nucleation is calculated.

Since we don't have observation data of ice nuclei and their properties including their chemical composition, we are not able to classify ice nuclei into contact nuclei, immersion and condensation nuclei, and deposition nuclei in a theoretical way. Hence, instead of obtaining $N_{a,cnt}$, $N_{a,imm}$, and $N_{a,dep}$ from the number concentrations of ice nuclei ($N_a$) theoretically based on the chemical composition of ice nuclei or instead of dividing $N_a$ into $N_{a,cnt}$, $N_{a,imm}$, and $N_{a,dep}$ theoretically based on the chemical composition of ice nuclei, in the simulations, to obtain individual $N_{a,cnt}$, $N_{a,imm}$, and $N_{a,dep}$ from $N_a$, we made a rough assumption that a third of $N_a$ is to be each of $N_{a,cnt}$, $N_{a,imm}$, and $N_{a,dep}$ and this assumption satifies $N_a = N_{a,cnt} + N_{a,imm} + N_{a,dep}$.

The averaged ice-crystal number concentration over grid points with non-zero ice-crystal number concentration at each time step is at the order of magnitude of 0.1 - 1 cm⁻³ in the runs. This range of the concentration is consistent with Lohmann and Diehl (2006) whose parameterizations are used for ice-crystal heterogeneous nucleation in this study as described above. In Figure 6 in Lohmann and Diehl (2006), for predicted CDNC around the order of magnitude of 100 cm⁻³, predicted ice-crystal number concentration varies between ~0.1 cm⁻³ and ~1 cm⁻³; note that as seen in our response to the comment on line 402 below, CDNC in the runs is approximately between 100 and 1000 cm⁻³. The consistency between this study and Lohmann and Diehl (2006) in terms of the range of ice-crystal number concentration indicates that ice-crystal number concentration in this study follows up well with the previous study whose ice-nucleation parameterizations are adopted by this study.

Line 241: What happens to aerosol from evaporating hydrometeors? If aerosol is recovered, information is lost, and I would not say that evolution is followed.

Aerosols in hydrometeors can come out of those hydrometeors and become present in the air when those hydrometeors containing those aerosols are entirely evaporated and disappear. In this study, this process is NOT considered. Instead, as described in text, at any grid points, immediately after clouds disappear entirely, aerosol size distributions and number concentrations recover to background properties that background aerosols at those points have before those points are included in clouds.

In this way, we can keep the concentrations of background aerosols outside clouds in the simulations at observed counterparts.

The corresponding text is revised as follows to remove confusion from the use of word "evolution":

(LL296-299 on p10-11)

In this way, we can keep concentrations of background aerosols outside clouds in the simulations at observed counterparts. This enables spatiotemporal distributions of background aerosols in the simulations to mimic those distributions that are observed and particularly associated with observed aerosol advection in reality.

Line 280: "However, background aerosol concentration acting as INP at each time step and grid point in the low-aerosol-s run is not different from that in the control-s run during the simulation period." This is a very surprising selection as throughout the manuscript the effect of aerosol on precipitation is discussed. Please change the wording and describe in more detail the temperature dependence of IN concentration.

In this study, we focus on effects of CCN, but not INP, on clouds and precipitation. This is because aerosols acting as CCN account for most of aerosol mass that affects clouds and precipitation, and those aerosols acting as CCN but not INPs are associated with major theories of aerosol-cloud interactions such as aerosol-induced invigoration of convection and intensification of gust fronts as described in "Introduction".

To clearly identify CCN effects on clouds and precipitation, we isolate CCN effects on clouds and precipitation by changing CCN concentrations only but not changing INP concentrations between the runs.  To clarify this point, text is revised or added as follows:

(LL54-57 on p3)

This study examines the role played by aerosols which act as cloud condensation nuclei (CCN) in the development of clouds and precipitation in two metropolitan areas in East Asia that have experienced substantial increases in aerosol concentrations over the last decades.

(LL110-117 on p4-5)

This study aims to examine effects of the increasing aerosols, which particularly act as cloud condensation nuclei (CCN), and their advection on clouds and precipitation in East Asia. This study focuses on aerosols which act as CCN, but not ice-nucleating

particles (INPs), to examine those effects, based on the fact that CCN account for most of aerosol mass that affects clouds and precipitation, and CCN, but not INPs, are associated with above-described aerosol-induced invigoration of convection and intensification of gust fronts. Note that these aerosol-induced invigoration and intensification are two well-established major theories of aerosol-cloud interactions.

(LL335-337 on p12)

However, to isolate CCN effects on clouds, background aerosol concentration acting as INPs at each time step and grid point in the low-aerosol-s run is not different from that in the control-s run during the simulation period.

In addition to revised text above, other words and phrases are revised to clearly indicate that this study is about CCN effects on clouds and precipitation but not INP effects. See manuscript for this revision.

There is no temperature dependence of IN concentration. As described in text, IN concentration is determined by observed aerosol properties at the surface (e.g., PM, aerosol size distribution and chemical composition), the assumption about the vertical distribution of aerosol concentrations and the assumption about the ratio of the IN concentration to the CCN concentration as detailed in text.

In authors' reply to the reviewer's comment about "the temperature dependence for heterogeneous nucleation", we describe how heterogeneous nucleation is parameterized by considering factors including the temperature dependence of heterogeneous nucleation.

Line 320: What is the total number of stations? Agreement with observations seems to be very good. Would it be possible to add the locations of stations into maps to see how well those cover the simulated area?

The total number of stations in Seoul (Beijing) area is ~300 (600). The locations of stations are marked in Figure 2. Displaying all stations in Figure 2 makes it messy. So, the locations of selected stations are displayed in Figure 2. These stations are selected in a way to represent overall distribution patterns of all stations well.  The marked locations in Figure 2 indicate that stations cover the simulation domain on land reasonably well.

343: What does "cumulative frequency distributions of precipitation rates" mean. How is rain rate cumulated. Also, more information is needed on the process used to extend the observations to fill the simulated domain.

**1. About cumulative frequency distribution of precipitation rates**

We classify precipitation rates at each time step and grid point in the whole domain and during the whole simulation period. The classified precipitation rates are put into corresponding bins of precipitation rates. Then, we count the number of those rates in each bin and the number in each bin is shown in Figure 6.

For example, let us assume that the minimum and maximum of precipitation rates over the whole domain and simulation period are 1 and 12, respectively, and we use a bin interval of 3. In this assumed situation, there are three bins. The first bin is between precipitation rates of 1 and that of 4, the second bin between 5 and 8 and the third bin between 9 and 12. Then, let us assume that there are 2 time steps during the whole simulation period and 2 grid points over the whole domain, and non-zero precipitation rates occur at all time steps and grid points. Then, there are 2 values of precipitation rates over the whole domain at each time step. Let us assume these values are as follows:

At the first time step:

   The first grid point: 1

    The second grid point: 7

At the second time step:

   The first grid point: 6

    The second grid point: 12

Then, there are "one count" of precipitation rate whose value is 1 for the first bin between 1 and 4, "two counts" of precipitation rates whose values are 6 and 7 for the second bin between 5 and 8, and "one count" of precipitation rate whose value is 12 for the third bin between 9 and 12. These "one count" or "two counts" in each bin correspond to "cumulative frequency of precipitation rates" in each bin. When the cumulative frequency in one of the bins is plotted together with that in the other bins as in Figure 6, this plot represents "cumulative frequency distribution of precipitation rates".

**2. About extrapolation/interpolation of observation data to grid points**

Observation data are extrapolated or interpolated into each time step and grid point in the simulations. For these extrapolation and interpolation, the inverse distance

weighting (IDW) method is used. The method is one of popular ones for the extrapolation and interpolation and detailed at https://en.wikipedia.org/wiki/Inverse_distance_weighting. To indicate the IDW method used, the following is added:

(LL222-224 on p8)

In this study, the inverse distance weighting method is used for the extrapolation and interpolation of observation data including aerosol mass into grid points and time steps in the model.

402: The number of cloud droplets formed should be presented to see how high concentrations can be found from the clouds to estimate the formation efficiency of warm precipitation.

The following is added:

(LL473-475 on p16)

CDNC, which is averaged over grid points and time steps with non-zero CDNC, is 1050 and 352 cm$^{-3}$ in the control-s and low-aerosol-s runs, respectively.

(LL634-636 on p21)

CDNC, which is averaged over grid points and time steps with non-zero CDNC, is 992 and 341 cm$^{-3}$ in the control-b and low-aerosol-b runs, respectively.

403: Are the differences in condensation rate meaningful to produce changes in updrafts? The increase in the condensation rate is five order of magnitude smaller than the change in liquid water content, thus being much longer than the lifetime of a single convective cell. Or is the main reason for increased liquid water the decrease in the precipitation formation efficiency which together with an increase in the updraft mass flux increases the amount of liquid water at higher altitudes also. This effect is not analyzed at all. I would expect the droplet concentration even in the low aerosol case to be high enough to maintain the supersaturation with respect to liquid water very close to zero.

We found errors in the program calculating condensation, deposition and freezing rates and due to these errors, just the order of magnitudes of those rates is wrong. These errors are related to the conversion between gram and kilogram. With the corrected order of magnitudes as shown in Figures 8 and 12 in the new manuscript, the increase in condensation rates is around two orders of magnitude smaller than the

change in liquid water content. Hence, it is hard to say that the time scale of the increase in condensation rates is longer than the lifetime of a single convective cell.

We repeated the control-s and low-aerosol-s runs by fixing CDNC and droplet size only for condensation which is related to drops (or droplets) whose sizes are smaller than 40 micrometers in radius. As mentioned in one of our responses above, in the microphysical scheme adopted in this study, for the calculation of cloud liquid content and rain water content, drops whose radius is smaller (greater) than 40 micrometers in radius are treated to be droplets (raindrops).

The fixed CDNC and droplet size are the average CDNC and size over grid points and time steps with non-zero CDNC in the low-aerosol-s run. The fixed CDNC and droplet size are applied to both of the repeated runs and thus there are no differences in CDNC and droplet size only for condensation between the repeated runs. The repeated control-s and low-aerosols-s runs are referred to as "the control-fixed-s run" and "the low-aerosol-fixed-s run". However, in these repeated runs, the predicted CDNC and sizes are applied to all of the other processes including collision and coalescence processes among drops or among raindrops or among raindrops and droplets. In these repeated runs, due to no aerosol-induced increases in CDNC and associated surface area of droplets for condensation, there are negligible aerosol-induced changes in condensation and updrafts as shown in supplementary Figure below for 12:00 LST when the MCS is at its mature stage.

There are aerosol-induced increases in cloud-liquid mass or content due to reduced autoconversion or decreases in the precipitation formation efficiency. These increases in cloud-liquid or droplet mass can induce more accretion of droplets by raindrops and thus precipitation by providing more droplets as a source of the accretion. However, there is an aerosol-induced decrease in precipitation between the repeated runs. The domain-averaged cumulative precipitation amount at the last time step is 11.2 mm and 12.3 mm in the control-fixed-s run and the low-aerosol-fixed-s run, respectively. This means that aerosol-induced decreases in precipitation formation efficiency and associated reduction in precipitation formation outweigh effects of autoconversion-reduction-induced increases in cloud-liquid mass on precipitation, leading to aerosol-induced reduction in precipitation, in the circumstances of negligible aerosol-induced changes in condensation and updrafts. This also means that there should be aerosol-induced increases in condensation, updrafts and associated increases in cloud liquid to induce aerosol-induced increases in precipitation by overcoming aerosol-induced reduction in the precipitation formation efficiency and associated reduction in precipitation formation, as seen in comparisons between the pair of the control-s and low-aerosol-s runs and that of the control-fixed-s and low-aerosol-fixed-s runs.

CDNC does not determine supersaturation solely. Environmental CAPE, which is controlled by a synoptic condition, and associated updrafts also affect supersaturation. For example, although CDNC is very high, there can be environmental CAPE and associated updraft speed that are high enough to maintain supersaturation with a significant magnitude. As mentioned in one of our responses above, the averaged CDNC over grid points and time steps with non-zero CDNC is 1050 and 352 cm$^{-3}$ in the control-s and low-aerosol-s runs.  Comparisons between the pair of the control-s and low-aerosol-s runs and that of the control-fixed-s and low-aerosol-fixed-s runs demonstrate that when the average CDNC changes from 352 cm$^{-3}$ in the low-aerosol-s run to 1050 cm$^{-3}$ in the control-s run, there are significant increases in condensation and associated updraft intensity. However, when the average CDNC is fixed at that in the low-aerosol run and does not vary from the low-aerosol-fixed-s run to the control-fixed-s run, there are negligible changes in condensation and updrafts. The time- and domain-averaged CAPE, basically determined by a given synoptic condition, is ~1000 J kg$^{-1}$ in the control-s, low-aerosol-s, control-fixed-s, and low-aerosol-fixed-s runs.  These four simulations indicate that a combination of CDNC, CAPE and associated updrafts in the low-aerosol-s run leads to a situation where supersaturation in the low-aerosol-s run is large enough to generate changes in condensation which are large enough to induce ~20 % changes in cumulative precipitation when CDNC increases from that in the low-aerosol-s run to that in the control-s run.

The control-s run is repeated again by increasing background concentrations of aerosols at each grid point and time step by a factor of 3.1; remember that the average background concentration of aerosols in the control-s run is higher than that in the low-aerosol-s run by a factor of 3.1. This repeated run is referred to as "the high-aerosol-s run". In the high-aerosol-s run, CDNC and droplet size are not fixed and predicted CDNC and size are applied to both condensation and all of the other microphysical processes including collision and coalescence processes. Increases in condensation and updraft intensity are ~ 2.5 times smaller on average from the control-s run to the high-aerosol-s run than from the low-aerosol-s run to the control-s run. This leads to a situation where there is only ~5% increase in precipitation amount from the control-s run to the high-aerosol-s. Remember that there is ~20% increase in precipitation amount from the low-aerosol-s run to the control-s run. This indicates that as the reviewer pointed out, lower supersaturation due to higher CDNC in the control-s run than in the low-aerosol run leads to a situation where aerosol-induced increases in condensation and updraft intensity are smaller from the control-s run to the high-aerosol-s run than from the low-aerosol-s run to the control-s run. This means that reducing supersaturation due to increasing CDNC lowers the sensitivity of condensation, updraft and precipitation to increasing aerosol concentrations, which is in line with the reviewer's argument. However, simulations here demonstrate that although supersaturation in the low-aerosol-s run can be considered low as the

reviewer pointed out, supersaturation in the low-aerosol-s run is at the magnitude with which increasing aerosol concentrations from the low-aerosol-s run to the control-s run can induce increases in condensation and updraft intensity significantly enough to in turn induce ~20% increases in precipitation.

Aerosol-induced increases in condensation and associated invigoration of updrafts in this study are consistent with the observational study of Koren et al. (2014) and the theoretical study of Igel and van den Heever (2021). Koren et al. (2014) and Igel and van den Heever (2021) showed that with more aerosol particles in clouds, there is more release of latent heat of condensation that induces invigoration of updrafts. In particular, Igel and van den Heever (2021) showed that in mixed-phase clouds, aerosol-induced increases in condensation play a more important role in the invigoration of updrafts than those in freezing.

References:

Koren, I., Dagan, G., & Altaratz, O. (2014). From aerosol-limited to invigoration of warm convective clouds. Science, 344(6188), 1143-1146.

Igel, A. L., & van den Heever, S. C. (2021). Invigoration or enervation of convective clouds by aerosols? Geophysical Research Letters, 48, e2021GL093804. https://doi.org/10.1029/2021GL093804

**Seoul case**
**(control-fixed-s run minus low-aerosol-fixed-s run)**

**12:00 LST**

[Figure]

Supplementary Figure. Vertical distributions of differences in the area-averaged condensation, deposition and freezing rates, and cloud-liquid and snow mass density, and updraft mass fluxes between the control-fixed-s and low-aerosol-fixed-s runs at 12:00 LST.

Line 422: It is surprising that differences in freezing rates are so small. Does it include also the ice/snow formed in collisions between liquid and frozen hydrometeors? Delayed warm precipitation formation should increase the liquid water content in mixed phase part of the cloud and thus provide more liquid for freezing also. Overall, can you estimate how big fraction of precipitation is formed through cold and warm processes and how it is changing with changing CCN concentration?

We found that we missed some of riming processes in the calculation of freezing rates. We re-calculated freezing rates by including all riming processes or collision processes between liquid and frozen (or solid) hydrometeors. These re-calculated freezing rates are reflected in Figures 8 and 12 and corresponding text in the new manuscript.

As described in text, CCN-induced increases in condensation in turn induce CCN-induced increases in updraft intensity. These increases in updraft intensity eventually induces CCN-induced increases in deposition for the Seoul case. For the Seoul case, over the whole simulation period and domain, differences in the average freezing rate between the control-s and low-aerosol-s runs are one to two orders of magnitude smaller than those in condensation and deposition rates as seen in Figure 8 in the new manuscript with the corrected freezing rates, although freezing rate increases with increasing CCN number concentrations. Hence, it is differences in condensation and deposition but not those in freezing that control differences in the mass of hydrometeors between the runs for the Seoul case.

It Is found that 61 (35) % of precipitable hydrometeors is generated by warm processes, while 39 (65) % of precipitable hydrometeors is generated by cold processes in the control-s (low-aerosol-s) run. We see that the ratio of generation of precipitable hydrometeors by warm processes to that by cold processes increases with increasing aerosol or CCN concentrations between the runs. Here, it should be emphasized that cold processes for each of the runs are predominantly controlled by deposition but not by freezing. The time- and domain-averaged freezing rate is ~one to ~two orders of magnitude smaller than deposition rate in each of the runs. This leads to above-mentioned negligible roles of freezing in differences in the mass of hydrometeors between the runs as compared to those roles of deposition for the Seoul case.

Line 658: Binned microphysics is employed, so there should not be a need for autoconversion as cloud droplet grow through coagulation coalescence to precipitation?

The corresponding text is revised as follows:

(LL766-771 on p26)

This and lower efficiency of collision and collection among droplets enable the control-b run to have a larger amount of cloud liquid or droplets as a source of evaporation. This in turn enables more droplet evaporation, more associated cooling and stronger downdrafts, although less rain evaporation is in the control-b run particularly for the period from 17:00 LST to 19:00 LST (Figure 17).

Lines 673-763: This is very interesting analysis and the change in the energy flow seems quite strong. The static energy itself is probably only slightly affected by aerosol (or is there more water in the PBL), so the change comes from the wind vector or how the averaging is done. As the selection of limiting value for areas A and B is quite arbitrary but still affecting the size of A and B areas, I would expect that the differences between different aerosol cases change when different criteria for areas A and B is employed. And thus, averaging affect the outcome. Maybe using net flux instead of domain averaged values could make this more convincing.

Here, we just want to emphasize that limiting values for aeras A and B are not arbitrary and based on Figure 15. We want to understand how the tipping precipitation rates of ~2 and ~12 mm hr$^{-1}$ between the control-b and low-aerosol-b runs are formed; see text for the details of the tipping precipitation rates. Based on Figure 15 and associated analysis, we find correspondence between the tipping precipitation rates and condensation rates. This correspondence indicates that condensation rates below ~3 × $10^{-3}$ g m$^{-3}$ s$^{-1}$ and above ~10 × $10^{-3}$ g m$^{-3}$ s$^{-1}$ are correlated with precipitation rates below ~2 mm hr$^{-1}$ and above ~12 mm hr$^{-1}$, respectively, while condensation rates between ~3 and ~10 × $10^{-3}$ g m$^{-3}$ s$^{-1}$ are correlated with precipitation rates between ~2 and ~12 mm hr$^{-1}$; see text for the details of this correspondence. As a way of understating the formation of the tipping precipitation rates, we are interested in the redistribution of precipitation between areas with precipitation whose rates are higher than ~12 mm hr$^{-1}$ and those with precipitation whose rates are between ~2 and ~12 mm hr$^{-1}$. To examine this redistribution, based on the correspondence, we examine areas (or area A) with condensation rates from 3×$10^{-3}$ g m$^{-3}$ s$^{-1}$ to 10 ×$10^{-3}$ g m$^{-3}$ s$^{-1}$ which correspond to precipitation whose rates are between ~2 and ~12 mm hr$^{-1}$ and areas (or area B) with condensation rates above 10×$10^{-3}$ g m$^{-3}$ s$^{-1}$ which correspond to precipitation whose rates are above ~12 mm hr$^{-1}$.

As shown in Figure 15, there is a standard deviation of the precipitation rate for each column-averaged condensation rate; note that to determine the above-described correspondence between the tipping precipitation rates and condensation rates, the mean precipitation rate for each condensation rate in Figure 15 is used. To check the robustness of results in Section e "moist static energy", this standard deviation and an associated range of precipitation rate (but not one mean precipitation rate) for each condensation rate are considered; here, the range extends over the standard deviation for each condensation rate and starts from the bottom of a vertical bar to the top of the bar in each condensation rate in Figure 15. Based on the range of precipitation rate, condensation rate corresponding to precipitation rates of ~12 mm hr$^{-1}$ (~2 mm hr$^{-1}$) can vary from ~9 ×$10^{-3}$ g m$^{-3}$ s$^{-1}$ to ~12×$10^{-3}$ g m$^{-3}$ s$^{-1}$ (from ~1 ×$10^{-3}$ g m$^{-3}$ s$^{-1}$ to ~4×$10^{-3}$ g m$^{-3}$ s$^{-1}$); here, the range of precipitation for each of condensation rates between ~9 ×$10^{-3}$ g m$^{-3}$ s$^{-1}$ and ~12×$10^{-3}$ g m$^{-3}$ s$^{-1}$ includes the precipitation rate of ~12 mm hr$^{-1}$,

while the range of precipitation for each of condensation rates between ~1 ×10$^{-3}$ g m$^{-3}$ s$^{-1}$ and ~4×10$^{-3}$ g m$^{-3}$ s$^{-1}$ includes the precipitation rate of ~2 mm hr$^{-1}$. We repeated the analysis in Section e by applying these variations of condensation rate to the determination of areas A and B. Here, based on the variations of condensation rate, we used condensation rate of each of 9, 10, 11, and 12 ×10$^{-3}$ g m$^{-3}$ s$^{-1}$ as the minimum (maximum) condensation rate for area B (area A) and condensation rate of each of 1, 2, 3 and 4 ×10$^{-3}$ g m$^{-3}$ s$^{-1}$ as minimum condensation rate for area A. Hence, there are 16 sets of the repeated analysis, considering the combination of 4 values of minimum (maximum) condensation rate for area B (area A) and 4 values of minimum condensation rate for area A. These sets demonstrate that results in Section e, which involve the net positive flux of the energy from area A to area B, are robust to the variation of condensation rates used to determine areas A and B.

In text, we describe that CCN-induced more condensation intensifies updrafts in area B, associated wind convergence and flow of moist static energy by wind convergence from area A to area B more in the control-b run than in the low-aerosol-b run. Hence, as the reviewer stated, changes in wind convergence or wind affects the flow of moist static energy. We want to reiterate that these changes in wind are triggered by changes in CCN concentrations and associated condensation. When we repeated the control-b and low-aerosol-b runs by removing latent heating by condensation in area B from 16:30 LST on, we find the absence of more net positive flux of the moist static energy from area A to area B in the repeated control-b run than in the repeated low-aerosol-b run, and no formation of the tipping precipitation rates in these repeated runs. These repeated runs demonstrate that CCN-induced differences in latent heating by condensation in area B trigger differences in wind convergence and associated flux or flow of the moist static energy from area A to area B between the control-b and low-aerosol-b runs, and form the tipping precipitation rates, as described in Section e.

Following the comment here and the other reviewer's comment, the total or net fluxes replace the domain-averaged fluxes in text. The corresponding text is revised accordingly.

Figure 8: Is it only snow here, or does it include all frozen hydrometeors? Also, the change in cloud fraction should be presented as all variables are presented area averaged. In addition, information related to temperature profile, at least the melting and heterogeneous freezing levels would help in understanding the changes at different altitudes.

In Figures 8 and 12, we added differences in the mass of other precipitable hydrometeors (i.e., raindrops and hail particles) between the runs. Accordingly, text is revised. Here, for the simplicity of the display, hail particles include graupel particles.

Also, want to note that snow mass in Figures 8 and 12 includes the ice-crystal mass for the simplicity of the display.

The melting and freezing levels are nearly at the level where temperature is 0 °C. The level with the temperature of 0 °C is marked in Figures 8 and 12 and accordingly, the following text is added:

(LL468-469 on p16)

In Figure 8, horizontal black lines represent the altitudes of freezing and melting.

(LL629-630 on p21)

In Figure 12, horizontal black lines represent the altitudes of freezing and melting.

Regarding the cloud fraction, the following is added:

(LL463-465 on p16)

Cloud fractions are 0.32 (0.30), 0.85 (0.82), 0.93 (0.92) and 1.00 (1.00) in the control-s (low-aerosol-s) run at 03:20, 03:40, 06:00 and 12:00 LST, respectively. We see that cloud fraction does not vary significantly between the runs.

(LL626-629 on p21)

Cloud fractions are 0.12 (0.11), 0.25 (0.22), 0.36 (0.32), 0.43 (0.40) and 0.48 (0.47) in the control-b (low-aerosol-b) run at 14:20, 15:40, 16:00, 17:20 and 19:00 LST, respectively. Here, we see that cloud fraction does not vary significantly between the runs.

To respond to a general comment above about changes in cloud-top heights with changing aerosols, the following is added. In the following, cloud depth, which is equivalent to cloud-top height, is presented.

(LL483-485 on p16-17)

This eventually leads to a situation where the maximum cloud depth is ~ 7 km in the control-s run and this depth is ~5 % deeper than that in the low-aerosol-s run for the whole simulation period.

(LL640-642 on p22)

These updrafts enable the maximum cloud depth to be ~ 12 km in the control-b run and this depth is just ~1 % deeper than that in the low-aerosol-b run for the whole simulation period.

Figure 9: This is complicated figure, and it is difficult to see what happens. Maybe the quality good be improved through decreasing the number of panels.

Figures 9c, 9d, 9g, 9h, 9i, 9j, 9m, 9n, 9q, 9r, 9w,9x,9y, and 9z are removed and text is revised accordingly.

---

## Author Response (AR2)

First of all, we appreciate the reviewer's comments and suggestions. In response to them, we have made relevant revisions to the manuscript. Listed below are our answers and the changes made to the manuscript according to those comments and suggestions. Each comment of the reviewer (in black) below is followed by our response (in blue).

Reviewer 1:

Authors want to note that in addition to dealing with comments below, we further removed unnecessary figures during this second revision process. We believe that this is in line with the reviewer's previous comment that suggested that we should remove unnecessary figures. Frankly speaking, the removal of those figures is partly motivated by our effort to reduce the publication fee. Since this paper is rather long, we need to reduce the fee by removing unnecessary figures and thus shortening the paper, considering our current budget. Although those figures are removed, there is no significant revision in associated text and there are no changes in the qualitative nature of the manuscript.

I feel that the authors have addressed the comments of the previous reviews.

However, I still have some confusion regarding the aerosol data used as input to the model. The authors state (p9, line 237) that only aerosol concentrations from the hour before the simulation starts are used. Observations during the simulation period were excluded, as AERONET sun photometer measurements are not available during cloudy periods. However, in Fig. 4, the authors show aerosol observations for the whole simulation period, and state in the caption that the background aerosol in the control-s simulation follows this temporal pattern. Lines 340-348 state that the simulated background aerosol concentrations change with time.

Are the authors also using aerosol data from national or municipal air quality monitoring stations? This would explain why PM10 was available in one city and PM2.5 was available in the other city, and why data was available during cloudy conditions. If so, the authors should properly cite the source of this data, and differentiate between what aerosol properties were derived from the AERONET data and what was derived from the other data sources.

More than 90% of the surface observation sites only measure up PM2.5 or PM10 (but not aerosol composition and size distribution) and are NOT a part of AERONET in the domains. These sites, which are not a part of AERONET, are owned and operated by the South-Korean or Chinese government. Just less than 10% of the surface observation sites in the domains are a part of AERONET. As stated in text, to represent aerosol composition and size distributions in the domains, data, which are from the AERONET sites and one hour before clouds start to form, are employed. Note that those PM2.5 or PM10 sites, which are NOT a part of AERONET BUT managed by the South-Korean or Chinese government, measure up aerosol mass using the beta-ray attenuation method that can measure up aerosol mass without relying on solar radiation and hence measure up aerosol mass continuously day and night regardless of the presence of solar radiation. Therefore, we obtain PM2.5 or PM10 data, which are from the observation sites managed by the South-Korean or Chinese government, throughout the simulation period in the domains. PM2.5 and PM10 in the data, which are from the sites managed by the South-Korean or Chinese government, vary spatiotemporally, since each of those sites measure its own aerosol mass at every observation time. Hence, PM2.5, which is from the sites managed by the South-Korean government and shown in Figure 4, changes with time.

As stated in text, it is assumed that for the whole domain and simulation period, aerosol composition and size distribution follow their counterparts derived using data from sites, which are a part of AERONET, for each of the Seoul and Beijing cases. As stated in text, with this assumption and using PM2.5 or PM10, which is not only produced by the sites managed by the South-Korean or Chinese government but also interpolated and extrapolated to grid points immediately above the surface and time steps, the background number concentrations of aerosols acting as CCN at those grid points immediately above the surface for simulations are obtained over the whole domain and simulation period for each of the cases.

To indicate the data sources, the following is added in "Code/data source and availability":

(LL983-984 on p33)

Note that in particular, the stored PM data are provided by the Korea Environment Cooperation in South Korea and State Key Laboratory of Severe Weather in China.

To clearly differentiate between what aerosol properties were derived from the AERONET data and what was derived from the other data, which are managed by the South-Korean government or Chinese government, the following is added:

(LL225-240 on p8-9)

There are surface observation sites, which measure aerosol properties, in the domains and these sites are classified into two types; the selected locations of these sites are marked by dots in the inner rectangles in Figure 1. The distance between the observation sites ranges from ~1 km to ~10 km and the time interval between observations is ~10 minutes. More than 90% of the sites belong to the first type of the sites. These first-type sites are managed by the government in South Korea or China, and measure $PM_{2.5}$ or $PM_{10}$ but not other aerosol properties such as aerosol composition and size distributions. Less than 10% of the sites belong to the second type of the sites. These second-type sites are a part of aerosol robotic network (AERONET; Holben et al., 2001) and measure aerosol composition and size distributions. The production of aerosol data in these second-type or AERONET sites is viable only in the presence of the sun. The first-type sites observe $PM_{2.5}$ or $PM_{10}$ using the beta-ray attenuation method (Eun et al., 2016; Ha et al., 2019) and hence, produce $PM_{2.5}$ or $PM_{10}$ data whether the sun is present or not. $PM_{2.5}/PM_{10}$ data from the first-type sites are used to represent the spatiotemporal variability of aerosols over the domains and the simulation periods. To represent aerosol composition and size distributions, data from the AERONET sites are employed.

(LL265-269 on p9-10)

By using $PM_{2.5}$ or $PM_{10}$, which is not only from the first-type sites but also interpolated and extrapolated to grid points immediately above the surface and time steps, and based on the assumption of aerosol composition and size distribution above, which is in turn based on data from the AERONET sites, the background number concentrations of aerosols acting as CCN are obtained for the simulation for each of the cases.

 Also, my understanding is that the current AERONET inversion algorithm (https://aeronet.gsfc.nasa.gov/new_web/Documents/Inversion_products_for_V3.pdf) is limited to estimating size distributions of aerosol in two lognormal modes with sizes above 0.05 μm, but the authors show in Fig. 3 size distributions that extend down to 0.01 μm with three lognormal modes.

In fact, we used the current AERONET inversion algorithm pointed out by the reviewer here to retrieve aerosol size distributions. These size distributions retrieved using the current AERONET algorithm were actually used for the calculation of aerosol concentrations which were used to get results displayed and discussed in the old manuscript.

The presented size distributions in Figure 3 in the old manuscript were not used to calculate aerosol number concentrations which were used to get results displayed and discussed in the old manuscript. We find that the nuclei modes with radius smaller than 0.05 micron in these distributions in Figure 3 in the old manuscript were created by measurement by an aircraft during an intensive observation period that was not associated with cases adopted by this study. Somehow, by error in the program code, this nuclei mode was combined with bi-modal aerosol size distributions which were based on the AERONET measurement in cases adopted by this study and retrieved by the current AERONET inversion algorithm. Resulting size distributions with this error were displayed in Figure 3 in the old manuscript. However, as mentioned, these distributions in the old manuscript were not used for results discussed and displayed in this study, and bi-modal distributions retrieved by the current AERONET inversion algorithm were actually used for the calculation of aerosol number concentrations which were used to get results displayed and discussed in the old manuscript. Accordingly, text is revised.

Here, we note that Figure 3 in the old manuscript is removed as a part of our effort to reduce the number of figures as we stated above. Figure 3 is removed, since in text, we give values of size distribution parameters (i.e., modal radii, standard deviations and the partition of aerosol number among modes), and we believe that this is enough to describe aerosol size distributions and showing them in a form of figure is redundant.

Technical corrections:

p7, line 188: 33 bins for each of how many size distributions? 5 (water, ice crystals, snow, graupel, hail)?

There are seven size distributions for hydrometeors that are classified into seven species. These seven species are water drops, snow aggregates, graupel, hail and three types of ice crystals which are plates, columns and dendrites. Each of seven species has its own size distribution and hence, there are seven size distributions for hydrometeors.

To clarify this, the corresponding text is revised as follows:

(LL181-185 on p7)

A set of kinetic equations is solved by the bin scheme to represent a size distribution function for each of seven classes of hydrometeors and aerosols acting as CCN. Hence, there are seven size distribution functions for hydrometeors. The seven classes of hydrometeors are water drops, three types of ice crystals, which are plates, columns and dendrites, snow aggregates, graupel and hail.

p8, lines 211-213: Please repeat the citation to Brown et al 2012 here.

Done.

p17, lines 496-498: "differences in freezing become at an order of magnitude, which is similar to that of differences in deposition, and become around one order of magnitude smaller than those in condensation" Do the authors mean "differences in freezing become similar in magnitude to differences in freezing and about one order of magnitude smaller than differences in condensation"?

Here, the difference in freezing is ~3 times smaller than that in deposition, while the difference in freezing is ~30 times smaller than that in condensation. Hence, in the old manuscript, we stated that the difference in freezing and that in deposition are at a similar order of magnitude, while stating that the difference in freezing is around one order of magnitude smaller than that in condensation. Regarding the difference in absolute magnitude, although the difference in freezing and that in deposition are at a similar order of magnitude, it is true that the difference in freezing is ~3 times smaller than that in deposition. Based on this, to remove confusion, text pointed out here is revised as follows:

(LL496-500 on p17)

After 06:00 LST until time reaches 12:00 LST when the overall differences in the cumulative precipitation frequency between the runs are established, differences in freezing become ~3 times smaller than those in deposition and ~one order of magnitude smaller than those in condensation (Figures 6c and 6d).

First of all, we appreciate the reviewer's comment. In response to it, we have made relevant revisions to the manuscript. Listed below are our answer and the changes made to the manuscript according to the comment. The comment of the reviewer (in black) is listed and followed by our response (in blue).

Reviewer 2:

My previous comments have been mainly addressed and corrections to the manuscript has been made accordingly. As a detail I would be still asking why the changes seen in cloud fraction are estimated to be insignificant? Up to 10% change does not sound small.

To remove confusion caused by the word "insignificant", the corresponding text is revised as follows:

(LL465-466 on p16)

We see that cloud fraction varies 0-~6% between the runs.

(LL623-624 on p21)

Here, we see that cloud fraction varies by ~2-12% between the runs.